# Estimating the glutamate transporter surface density in distinct sub-cellular compartments of mouse hippocampal astrocytes

Anca R. Rădulescu[1]*, Gabrielle C. Todd[2], Cassandra L. Williams[1], Benjamin A. Bennink[2], Alex A. Lemus[2], Haley E. Chesbro[2], Justin R. Bourgeois[3], Ashley M. Kopec[3], Damian G. Zuloaga[4], Annalisa Scimemi[2]*

**1** Department of Mathematics, State University of New York at New Paltz, New Paltz, New York, United States of America, **2** Department of Biology, State University of New York at Albany, Albany, New York, United States of America, **3** Department of Neuroscience and Experimental Therapeutics, Albany Medical College, Albany, New York, United States of America, **4** Department of Psychology, State University of New York at Albany, Albany, New York, United States of America

* radulesa@newpaltz.edu (ARR); scimemia@gmail.com (AS)

**Data Availability Statement:** The data included in this manuscript are available on Open Science Framework (https://osf.io/bv29c/). The code for running the 3D Monte Carlo reaction-diffusion

## Abstract

Glutamate transporters preserve the spatial specificity of synaptic transmission by limiting glutamate diffusion away from the synaptic cleft, and prevent excitotoxicity by keeping the extracellular concentration of glutamate at low nanomolar levels. Glutamate transporters are abundantly expressed in astrocytes, and previous estimates have been obtained about their surface expression in astrocytes of the rat hippocampus and cerebellum. Analogous estimates for the mouse hippocampus are currently not available. In this work, we derive the surface density of astrocytic glutamate transporters in mice of different ages via quantitative dot blot. We find that the surface density of glial glutamate transporters is similar in 7-8 week old mice and rats. In mice, the levels of glutamate transporters increase until about 6 months of age and then begin to decline slowly. Our data, obtained from a combination of experimental and modeling approaches, point to the existence of stark differences in the density of expression of glutamate transporters across different sub-cellular compartments, indicating that the extent to which astrocytes limit extrasynaptic glutamate diffusion depends not only on their level of synaptic coverage, but also on the identity of the astrocyte compartment in contact with the synapse. Together, these findings provide information on how heterogeneity in the spatial distribution of glutamate transporters in the plasma membrane of hippocampal astrocytes my alter glutamate receptor activation out of the synaptic cleft.

## Author summary

In this work, we use experimental and modeling approaches to estimate the surface density of the glial glutamate transporters GLAST and GLT-1 in mouse hippocampal astrocytes. The results show that the surface density of glutamate transporters in hippocampal astrocytes is similar in 7–8 week old mice and rats. In mice, GLAST and GLT-1 reach their peak expression at ∼ 6 months of age. Their expression, however, varies widely

simulations with CellBlender and for generating the 3D geometry of *in silico* astrocytes can be found at https://github.com/scimemia/Glutamate-transporters-estimates.

**Funding:** This work was supported by the Simons Foundation Collaboration Grants for Mathematicians (523763; A.R.), the SUNY New Paltz Research Scholarship and Creative Activities (A.R., C.L.W.), Albany Medical College start-up funds (A.M.K.), the National Institutes of Health (R03-AG07011 to A.M.K., R15-MH118692 to D.G. Z., R03-NS102822 to A.S.), and National Science Foundation (IOS1655365, IOS2011998 to A.S.). The funders had no role in study design, data collection and analysis, decision to publish, or preparation of the manuscript.

**Competing interests:** The authors have declared that no competing interests exist.

across different sub-cellular compartments. These findings indicate that the identity of the sub-cellular compartment of an astrocyte in contact with a glutamatergic synapses, in addition to the extent of its synaptic coverage, is a main factor to limit glutamate spillover and receptor activation at a distance from the release site.

## Introduction

Glutamate removal from the extracellular space relies on passive diffusion and active transport, a phenomenon mediated by transporter molecules abundantly expressed in the plasma membrane of astrocytes [1–3]. In the rat hippocampus, $\sim 57\%$ of excitatory synapses are contacted by astrocytic membranes, which surround $\sim 43\%$ of the synaptic perimeter [4]. Structural, electrophysiological and modeling studies indicate that small changes in the density of expression of glutamate transporters can profoundly change the lifetime of glutamate in the extracellular space, receptor activation, and the time course of excitatory synaptic currents [5–7]. For this reason, it is important to have detailed understanding of how their surface density varies across species and across different domains of the astrocyte plasma membrane.

The first attempts to estimate the surface density of glutamate transporters in astrocytes was based on electrophysiological studies in the hippocampus and cerebellum. These studies suggested that glutamate transporters are at least two orders of magnitude more abundant than glutamate receptors, but the range of these estimates was very broad (1,315–13,150 $\mu m^{-2}$) [8–10]. An alternative approach, which we summarize here, relied on quantitative immuno-blotting of protein extracts [11]. *First*, the immunoreactivity of GLAST and GLT-1 in whole tissue extracts was compared with the immunoreactivty of purified protein standards isolated with an immunoaffinity purification protocol. This, in the adult rat hippocampus, led to density estimates of 3,200 $\mu m^{-3}$ for GLAST and 12,000 $\mu m^{-3}$ for GLT-1. *Second*, electron microscopy (EM) sections were used to identify astrocyte profiles, and calculate the linear density of astrocytic membranes to extrapolate their corresponding 3D values. These astrocyte membrane density values ($\sim 1.3 - 1.6$ $\mu m^2/\mu m^3$) were collected from layers I-IV of the visual cortex of P23–25 male Long Evans rats [12]), but were used to derive estimates for the rat hippocampus and cerebellum. *Third*, the astrocyte membrane density values were used to convert glutamate transporter concentrations into measures of glutamate transporter density in astrocytic membranes (2,300 $\mu m^{-2}$ and 8,500 $\mu m^{-2}$ molecules for GLAST and GLT-1, respectively [11]). Note that for these calculations, it was assumed that GLAST and GLT-1 are only expressed in astrocytes, where these transporters are known to be most abundant [13, 14]. This assumption appears to be reasonable for the hippocampus, where the protein expression of GLAST is limited to astrocytes [14–16], and only very small amounts of the GLT-1 protein have been detected in neurons [1, 17–26]. Accordingly, neuronal GLT-1 accounts for only 5 – 10% of all GLT-1 molecules [20], and its physiological role remains a mystery, with data suggesting that selective neuronal deletion of GLT-1 does not lead to obvious behavioral consequences or survival [27, 28]. Clearly, the approximation of GLAST/GLT-1 being purely astrocytic proteins is not meant to dismiss a potentially important and only partially understood functional relevance of neuronal GLT-1 [19, 29, 30]. However, in the context of this work, given the relatively small expression levels of GLT-1 in neurons, the approximation of it being only localized in astrocytic membranes is unlikely to introduce major distortions in the surface density estimates for this transporter [1, 18, 20]. The calculations described above also assumed that GLAST/GLT-1 are only present in the plasma membrane, and not in the membrane of intracellular organelles. Even in this case, the approximation is not meant to be perfect, but it is

reasonable because the proportion of transporters expressed in intracellular membranes like the endoplasmic reticulum is small compared to the plasma membrane, with likely negligible effects on the final surface density estimates [1, 11, 16]. With these considerations in mind, the average surface density of all glutamate transporters in astrocytic membranes of the rat hippocampus have settled to a total of $\sim 10,800 \, \mu m^{-2}$ and these measures appeared consistent with indirect estimates collected using electrophysiological recordings [1, 9, 11, 31].

These values of GLAST/GLT-1 surface density have been used extensively in the scientific community, particularly in computational models of glutamate clearance from the extracellular space. Despite having been derived from rats, and despite the fact that some key parameters were obtained from different brain regions and in rats of different ages (e.g., the astrocyte membrane density values), these estimates have served as reference values for modeling glutamate uptake in other rodent species (e.g., mice) of different ages [6, 7, 32–39]. Whether this is a legitimate assumption, and whether the estimates of [11] hold true for hippocampal astrocytes of mice of different ages, which are now widely used in the scientific community, remains unclear.

Another aspect that remains incompletely understood is whether it is fair to assume that the distribution of glutamate transporters is homogeneous throughout the astrocyte plasma membrane. Shortly after the first isolation and reconstitution of glutamate transporters, immunolabeling light microscopy studies suggested the existence of a higher density of glutamate transporters in small astrocytic processes, and lower expression levels in astrocyte cell bodies [13]. These findings were reproduced by other groups using immunolabeling EM and confocal microscopy in acute and organotypic hippocampal slice cultures [16, 40–42]. More recently, the existence of a clustered distribution of GLT-1 in the plasma membrane of astrocytes has been confirmed using super-resolution STORM and STED multi-color microscopy imaging [43, 44]. Some of the GLT-1 clusters remain stable over time, whereas others appear to undergo remodeling [42]. Under basal conditions, the mobile GLT-1 molecules have a median instantaneous diffusion coefficient of $2.1 - 2.3 \cdot 10^{-2} \, \mu m^2/s$ at 37°C [2, 45], which is similar to that of mobile GluN2B-containing NMDA receptors ($2.5 \cdot 10^{-2} \, \mu m^2/s$ [46]), but much higher than that of mobile AMPA ($7.5 \cdot 10^{-3} \, \mu m^2/s$ [47]) and GluN2A-containing NMDA receptors ($7.5 \cdot 10^{-4} \, \mu m^2/s$ [46]). The diffusion coefficient of GLT-1 is increased in astrocyte cell bodies and decreased upon glutamate exposure, suggesting that there is an activity-dependent confinement of GLT-1 around excitatory synapses [2, 3, 45]. In addition to being clustered (i.e., confined to hot spots), the distribution of glutamate transporters along the cell membrane appears to be highly non-uniform. Accordingly, the expression of GLAST in fine astrocytic processes is roughly half of that at the soma, whereas the expression of GLT-1 in astrocyte cell bodies is sparse [44, 48]. Molecular studies suggest that an extracellular loop within the $4^{th}$ transmembrane region of the GLT-1 protein is responsible for its trafficking and localization in small astrocyte processes referred to as tips or leaves [48].

While these findings provide evidence for the existence of a heterogeneous expression of glutamate transporters along the astrocyte membrane, a number of questions remain currently unanswered. How does the local surface density of glutamate transporters change in different sub-cellular domains of the astrocyte membrane? How do geometrical constraints limit the local density of these molecules in astrocyte tips and leaves, the smallest membrane processes formed by these cells? Here, we address these questions using a combination of experimental measures of GLAST/GLT-1 expression in the hippocampus of mice aged from 2 weeks to 21 months, and theoretical models. We analyze the effect of geometrical constraints on the local surface density of glutamate transporters. Our data show that the astrocyte tips and leaves, which contain the highest surface density of these molecules, represent only a small proportion of the entire astrocytic membrane. This suggests that many hippocampal synapses may be

contacted by other sub-cellular compartments of the astrocyte membrane, with lower transporter expression. We analyze how these changes alter the activation of glutamate receptors at a range of distance from the release site, and with release sites located closer or farther away from neighboring glial processes.

## Materials and methods

### Ethics statement

All experimental procedures were performed in accordance with protocols approved by the Institutional Animal Care and Use Committee at SUNY Albany, Albany Medical College, and guidelines described in the National Institutes of Health's Guide for the Care and Use of Laboratory Animals.

### Mice

Unless otherwise stated, all mice were group housed and kept under 12 hour:12 hour light: dark conditions (lights on at 7 AM; lights off at 7 PM), with food and water available *ad libitum*. All experiments were performed on C57BL/6NCrl wild-type mice of either sex, aged from 2 weeks to 21 months.

### Rats

Unless otherwise stated, all rats were group housed and kept under 12 hour:12 hour light:dark conditions (lights on at 7 AM; lights off at 7 PM), with food and water available *ad libitum*. All experiments were performed on Sprague Dawley male rats purchased from (Boyertown, PA) at P21, and aged 7–8 weeks.

### Dot blotting

Dot blot experiments were performed on protein extracts from the hippocampus of mice of either sex aged 2 weeks to 21 months. Membrane proteins were extracted using the Mem-PER Plus Membrane Protein Extraction Kit according to the manufacturer's instructions (Cat# 89842; ThermoFisher Scientific, Waltham, MA), using a mixture of protease and phosphatase inhibitors (10 $\mu$l/ml, Cat# 78441; ThermoFisher Scientific, Waltham, MA). Cross-contamination of cytosolic proteins into the membrane fraction is usually less than 10%. The total membrane protein concentration was determined using a Bradford assay (Cat# 5000006, BioRad, Hercules, CA). Purified, GST-tagged, recombinant protein of the human homologs of GLAST and GLT-1 (*Slc1a3*, Cat# H00006507-P02; *Slc1a2*, Cat# H00006506-P01; Abnova, Taipei, Taiwan) were serially diluted in phosphate buffered saline (PBS) and used to generate standard curves. Purified, recombinant protein and total hippocampal membrane protein were spotted on polyvinylidene difluoride (PVDF) membranes (Cat# P2563; MilliporeSigma, Burlington, MA). The membranes were blocked with 5% nonfat milk in Tris-buffered saline with 0.1% Tween 20 (TBST) pH 7.6, and probed with primary antibodies (1 : 1, 000 for GLAST and GLT-1) for 1–2 hours at room temperature in 5% bovine serum albumin (BSA) in TBST, pH 7.6. The blots were subsequently incubated with biotinylated horse anti-rabbit antibody (1:1,000) for 1 hour at room temperature in 5% BSA in TBST, pH 7.6. We amplified the immuno-labeling reactions using the Vectastain ABC kit (PK-6100, Vector Laboratories, Burlingame, CA) (1 : 2, 000 for GLAST and GLT-1), and the Clarity Western enhanced chemiluminescence (ECL) system served as the substrate for the peroxidase enzyme (Cat# 1705060, Bio-Rad, Hercules, CA). A list of antibodies and reagents used for the dot blot experiments, together with their RRID, is reported in Table 1.

**Table 1. List of antibodies and reagents used for dot blot experiments.** The table lists the RRID number of antibodies and reagents used for dot blot analysis.

| RRID | Item | Supplier |
|---|---|---|
| AB_955879 | rabbit anti GLAST | Abcam, Cambridge, MA |
| AB_11005463 | rabbit anti GLT-1 | Novus, Littleton, CO |
| AB_2336201 | biotinylated horse anti-rabbit | Vector Laboratories, Burlingame, CA |
| AB_2336819 | Vectastain ABC kit | Vector Laboratories, Burlingame, CA |

For our quantitative analysis, protein dot intensities were collected as 16-bit images using a digital chemiluminescence imaging system using different exposure times (1 s – 10 min; c300, Azure Biosystems, Dublin, CA). Each image was converted to an 8-bit image for analysis using the Fiji software (https://fiji.sc/). Only images collected at exposure times that did not lead to saturation were included in the analysis. The intensity of each dot was calculated as the RID in a region of interest (ROI) drawn around each dot. Background correction was carried out for each dot by subtracting the average RID of three dots containing only PBS. Each concentration of purified protein or total hippocampal membrane protein was spotted 2–3 times (1 $\mu$l each). We averaged the RID values after background correction. Each age group contained protein samples from n = 3 mice. The molecular weights of the human, GST-tagged, recombinant proteins (87,387 Da and 89,988 Da for GLAST and GLT-1, respectively) and of the endogenous mouse proteins (59,619 Da and 62,027 Da for GLAST and GLT-1 in mice, and 54,374 Da and 60,914 Da for GLAST and GLT-1 in rats, respectively) were calculated based on their amino acid sequences using the software BioEdit (https://bioedit.software.informer.com/7.2/). We also accounted for the purity of the recombinant proteins (92.82% for GLAST and 86.93% for GLT-1). Mass corrected standard curves correlating average RID in each ROI to the amount of purified GLAST or GLT-1 spotted onto the membrane were fitted using the following sigmoidal curve:

$$F(x) = \frac{max}{1 + exp\left(\frac{x_{half} - x}{rate}\right)} \tag{1}$$

Based on this fit, we calculated the fraction of GLAST and GLT-1 in dots from total hippocampal membrane proteins. We assumed that there are 97.8 g protein per Kg of hippocampal tissue [11], and that the density of brain tissue is 1.05 $g/cm^3$ [11]. Furthermore, we calculated the surface density of astrocytic membrane in *stratum radiatum* of the mouse hippocampus using 3D axial STEM tomography reconstructions (see below). Based on this analysis, the estimated astroglial surface density is 3.15 $\mu$m$^2$/$\mu$m$^3$. With these assumptions, we were able to estimate the average number of GLAST and GLT-1 molecules/$\mu$m$^2$ on the membrane surface of hippocampal astrocytes in tissue samples from different age groups: from 2 weeks to 21 month old mice. Note that a small proportion of GLT-1, the most abundant glial glutamate transporter in the adult mammalian brain, is also expressed in neurons (5–10%) [18]. For consistency with [11, 16], we assumed that all the GLAST/GLT-1 molecules isolated in our dot blot experiments belonged to astrocytes did not account for this in our surface density calculations. Likewise, we did not account for any potential expression of GLAST/GLT-1 in oligodendrocytes, which has been detected in regions of the brain other than the hippocampus [23].

## EM and 3D axial STEM tomography

Acute hippocampal slices processed for EM analysis were prepared as described for the electrophysiology experiments, from P17 mice. Slices were microwave-fixed for 13 s in 6%

glutaraldehyde, 2% PFA, 2 mM $CaCl_2$ in 0.1 N sodium cacodylate buffer and stored overnight at 4˚C. After three washes in 0.1 N cacodylate buffer, we cut samples from the middle part of CA1 *stratum radiatum*, $\sim$ 100 $\mu$m away from the pyramidal cell layer. These samples were treated with 1% osmium tetroxide for 1 hour on ice, *en bloc* mordanted with 0.25% uranyl acetate at 4˚C overnight, washed and dehydrated with a graded series of ethanol, and embedded in epoxy resins. Thin sections (70–90 nm) were counter-stained with lead citrate and uranyl acetate and examined on a JEOL 1200 EX transmission electron microscope. Images were collected with a CCD digital camera system (XR-100, AMT, Woburn, MA). To visualize the arrangement of post-synaptic densoties (PSDs) and astrocytic processes, thick sections ($\sim$ 1 – 1.5 $\mu$m) were cut from regions of CA1 *stratum radiatum* and electron tomograms were collected in a 300 kV electron microscope operated in the scanning transmission EM (STEM) mode, as described previously [49, 50]. A sample thickness of 1 $\mu m$ (enabled by axial STEM tomography) provided sufficient sample depth to visualize features of interest in their entirety, such as synapses. In contrast to STEM tomography, conventional TEM tomography is limited to a specimen thickness of $\sim$ 400 nm and cannot be applied to such thick sections because the transmitted electrons undergo multiple inelastic scattering processes, resulting in images that are blurred by chromatic aberration of the objective lens. Axial STEM tomography is not affected by chromatic aberration because the objective lens that forms the electron probe is in front of the specimen. Dual-axis tilt series of selected sections were recorded using an FEI Tecnai TF30 TEM/STEM operating at 300 kV (1.5˚ tilt increment, tilt range from 55˚ to -55˚, pixel size = 5.6 nm). Image registration, tomogram generation, tracing, surface area and volume measures were performed using IMOD 4.7 (https://bio3d.colorado.edu/imod/). In the tomograms, we identified astrocytic processes based on their shape and cytoplasmic structure (i.e., presence of intermediate filaments and a relatively clear cytoplasm), their lack of synaptic vesicles and post-synaptic densities and because they did not give rise to pre- or post-synaptic terminals [4]. Orthoslices through the STEM tomograms showed that astrocytic processes contained glycogen granules, intermediate filament bundles and a more electron-lucent cytoplasm with respect to that of neurons. The astrocytic processes were traced for the entire thickness of the reconstructed volume (1.5 $\mu$m). We reconstructed all astrocytic processes in a tissue block.

## Acute slice preparation

Acute coronal slices of the mouse hippocampus were obtained from C57BL/6NCrl mice of either sex (P16–21), deeply anesthetized with isoflurane and decapitated in accordance with SUNY Albany Animal Care and Use Committee guidelines. The brain was rapidly removed and placed in ice-cold slicing solution bubbled with 95% $O_2$/5% $CO_2$ containing the following (in mM): 119 NaCl, 2.5 KCl, 0.5 $CaCl_2$, 1.3 $MgSO_4 \cdot H_2O$, 4 $MgCl_2$, 26.2 $NaHCO_3$, 1 $NaH_2PO_4$, and 22 glucose, 320 mOsm, pH 7.4. The slices (250 $\mu$m thick) were prepared using a vibrating blade microtome (VT1200S; Leica Microsystems, Buffalo Grove, IL). Once prepared, the slices were stored in slicing solution in a submersion chamber at 36˚C for 30 min and at room temperature for at least 30 min and up to 5 hours. Unless otherwise stated, the recording solution contained the following (in mM): 119 NaCl, 2.5 KCl, 2.5 $CaCl_2$, 1 $MgCl_2$, 26.2 $NaHCO_3$, and 1 $NaH_2PO_4$, 22 glucose, 300 mOsm, pH 7.4. We identified the hippocampus under bright field illumination and astrocytes using differential interference contrast, using an upright fixed-stage microscope (BX51 WI; Olympus Corporation, Center Valley, PA). Whole-cell, voltage-clamp patch-clamp recordings were made using patch pipettes containing (in mM): 120 $CsCH_3SO_3$, 10 EGTA, 20 HEPES, 2 MgATP, 0.2 NaGTP, 5 QX-314Br, 290 mOsm, pH 7.2. Biocytin 0.2–0.4% was added to the internal solution. All recordings were

obtained using a Multiclamp 700B amplifier (Molecular Devices, San Jose, CA), filtered at 10 KHz, converted with an 18-bit 200 KHz A/D board (HEKA Instrument, Holliston, MA), digitized at 10 KHz, and analyzed offline with custom-made software (A.S.) written in IgorPro 6.37 (Wavemetrics, Lake Oswego, OR). Patch electrodes (#0010 glass; Harvard Apparatus, Holliston, MA) had tip resistances of $\sim$5 M$\Omega$. Series resistance ($\sim$20 M$\Omega$) was not compensated, but was continuously monitored and experiments were discarded if this changed by more than 20%. All reagents were purchased from Millipore Sigma (Burlington, MA). All recordings were performed at room temperature.

### Biocytin fills, proExM and 2P-LSM image acquisition

Patched astrocytes were maintained in the whole-cell configuration and voltage-clamp mode, at a holding potential of -90 mV for at least 30 min, to allow biocytin to fill small cellular compartments. At the end of this time, the patch pipette was gently removed and the slice was fixed with PBS 4% PFA overnight at 4˚C, cryoprotected in PBS 30% sucrose and stored in PBS for up to 1 week. Each slice was then incubated in 0.1% streptavidin-Alexa Fluor 488 conjugate and 0.1% Triton X-100 for 3 hours at room temperature. The day before starting the protein retention expansion microscopy (proExM) protocol [51], we incubated the slice with DAPI Fluoromount G overnight at 4˚C (Cat# 0100–20; SouthernBiotech, Birmingham, AL) to ensure the whole slice could be visualized under fluorescence microscopy to localize the biocytin-filled, streptavidin-labelled astrocyte. The proExM protocol, originally developed by [51], was analogous to the one used in our previous work [52]. Briefly, each slice was incubated with 200 $\mu$l of anchoring solution overnight at room temperature. The following day, the slices were gelled and digested with Proteinase K overnight at room temperature. Subsequently, they were expanded using three consecutive incubations with distilled water for 15 min each. The expanded gels containing biocytin-filled astrocytes were then covered with 2% agarose and submerged in distilled water before being imaged with a custom-made two-photon laser-scanning microscope. The two-photon laser scanning system (Scientifica, Clarksburg, NJ) was powered by a Ti:sapphire pulsed laser (Coherent, Santa Clara, CA) tuned to 760 nm and connected to an upright microscope with a 60×/1.0 NA objective (Olympus Corporation, Center Valley, PA). The green fluorescent signal was separated from the red using a 565 nm dichroic mirror and filtered using FITC filter sets (Olympus Corporation, Center Valley, PA). Under these experimental conditions, the optical lateral resolution was $res_{x,y}$ = λ/2NA = 380 nm and the axial resolution was $res_z$ = 2λ/NA$^2$ = 1.52 $\mu$m. We averaged eight frames for each optical section (260×260 $\mu$m, 512×512 pixels, pixel size: 0.51 $\mu$m) and collected z-stacks for a total distance of $\sim$200 $\mu$m (in 1.20 $\mu$m steps). The composition of the anchoring, gelling and digestion solution are reported in Table 2. Note that the linear expansion factor of proExM in our experimental conditions was $\sim$2 [52]. Taking this correction into account, our effective optical resolution was 0.19 $\mu$m × 0.19 $\mu$m × 0.76 $\mu$m, and our voxel size was 0.26 $\mu$m × 0.26 $\mu$m × 0.60 $\mu$m.

### Analysis of 2P-LSM images of proExM treated tissue

Adjacent z-stack tiles acquired using 2P-LSM were stitched using the Pairwise Stitching plugin available in Fiji (https://imagej.net/Fiji), with Linear Blending Fusion Method (check peaks value: 95). After stitching, we imported the z-stacks into Bitplane Imaris 9.3.1 (Oxford Instruments, Abingdon, UK) and converted them to 8-bit .ims files. The next steps aimed to isolate the biocytin-filled astrocyte from other gap-junction coupled astrocytes, typically much dimmer. *First*, we smoothed each image using a Gaussian filter with a width of 0.51 $\mu$m, matching the *x,y* pixel size of the image stacks collected using 2P-LSM. *Second*, we generated a surface

**Table 2. Composition of anchoring, gelling and digestion solution for proExM.** The tables list the chemical reagents that were used to process hippocampal slices for proExM.

| Anchoring solution (200 $\mu l$/slice) | Cat# | Supplier | Amount |
|---|---|---|---|
| Acryloyl X, SE | A20770 | Life Technologies | 0.1 mg/ml |
| DMSO | 472301 | Millipore Sigma | 2 $\mu l$ |
| PBS | P4417 | Millipore Sigma | 198 $\mu l$ |
| **Monomer solution (188 $\mu l$/slice)** | **Cat#** | **Supplier** | **Amount** |
| Sodium acrylate | 408220 | Millipore Sigma | 86 mg/ml |
| Acrylamide | A9099 | Millipore Sigma | 25 mg/ml |
| N,N'-methylenebisacrylamide | M7279 | Millipore Sigma | 1.5 mg/ml |
| NaCl | 215700 | BeanTwon Chemical | 117 mg/ml |
| PBS 10X | P4417 | Millipore Sigma | 18.8 $\mu l$ (1:10) |
| Distilled water | | | 169.2 $\mu l$ |
| **Gelling solution (200 $\mu l$/slice)** | **Cat#** | **Supplier** | **Amount** |
| Monomer solution | | | 188 $\mu l$ |
| 4-hydroxy-TEMPO | 176141 | Millipore Sigma | 4 $\mu l$ (0.01%) |
| TEMED | T9281 | Millipore Sigma | 4 $\mu l$ (0.2% w/w) |
| APS | A3678 | Millipore Sigma | 4 $\mu l$ (0.2% w/w) |
| **Digestion solution (200 $\mu l$/slice)** | **Cat#** | **Supplier** | **Amount** |
| Tris pH 8.0 | AM9855G | Ambion | 50 mM |
| EDTA | 03690 | Fluka Analytical | 1 mM |
| Triton X-100 | X100 | Millipore Sigma | 0.5% v/v |
| Guanidine hydrochloride | G3272 | Millipore Sigma | 0.8 M |
| Proteinase K | P6556 | Millipore Sigma | 8 U/ml |

mask using an absolute intensity threshold that allowed us to get rid of all voxels with an intensity value greater than 20. *Third*, we segmented the surface mask using a seeded region-growing method using a seed point diameter of 20 $\mu$m and a quality threshold of 10. *Fourth*, we manually deleted surfaces outside the patched astrocyte. This allowed us to refine the surface mask to be used in the next step. *Fifth*, we applied this refined surface mask to the original z-stack. This allowed us to isolate voxels belonging to the astrocyte of interest based on their location (i.e., they resided within the mask) and gray intensity (i.e., gray intensity >1). All other voxels outside this mask were assigned an intensity value of zero. To generate the cytoplasmic volume of the astrocyte, in Bitplane Imaris, we selected the channel created using a gray intensity >1, applied the seeded region growing algorithm (seed point diameter: 20 $\mu$m) and manually set an arbitrary quality threshold that allowed us to identify only one surface within the astrocyte (typically between 10 and 20). No surface smoothing was applied. To complete the surface rendering, Bitplane Imaris uses a marching cubes computer graphics algorithm. Briefly, the marching cubes algorithm allows to determine whether an arbitrary set of points is within the boundary of an object. To do this, the volume occupied by these points is divided into an arbitrary number of cubes and the algorithm tests whether the corners of those cubes lie within the selected object. All cubes where corners are found both within and outside of the object produce a set of points that define the boundary of a surface that identifies the cytoplasmic volume [53]. We used a similar approach to generate the bounding volume. In this case, we applied a surface smoothing factor of 5 $\mu$m, meaning that structural details smaller than 5 $\mu$m (2.5 $\mu$m without proExM) were lost. To generate filaments, which represent astrocyte branches of different levels, we selected the masked channel containing only the biocytin-filled astrocyte, and used the Autopath (no loops) filament tracing algorithm for big data

sets. The Autopath (no loops) algorithm produces a tree-like filament (based on local intensity contrast) that connects large (25.4 $\mu$m at the soma) to small seed points (1 $\mu$m at the most distal processes). The origin of the primary branches was set to the center of the soma. For the filament branch count, we masked the portion of the primary branches that fell within the radius of the soma. We calculated branch diameters from input parameters rather than from the image intensity values.

## 3D Monte Carlo reaction-diffusion simulations

We used a 3D, realistic Monte Carlo reaction-diffusion model to analyze the diffusion properties of glutamate in and out of the cleft. This model allowed us to implement stochastic reaction-diffusion in a 3D environment, and incorporates the major sources of variability that can affect glutamate diffusion, and the 3D geometry of adjacent pre- and post-synaptic terminals. We analyzed the dynamics of glutamate diffusion and receptor activation in a 3D environment that contained a central synapse (from which glutamate was released) surrounded by six inactive synapses. In this regular arrangement, all synapses were 500 nm apart [5]. The geometry of seven adjacent synaptic contacts was created *in silico* with an open source program (Blender 2.79b) and an add-on that allows creating computational models for use in a Monte Carlo simulation environment (MCell) and visualizing the simulation results (CellBlender 3.5.1.1). The simulations were run within a cube (world), with transparent properties for all diffusing molecules (3 $\mu$m $\times$ 3 $\mu$m $\times$ 3 $\mu$m). Each pre- and post-synaptic terminal was modelled as a hemisphere with radius of 250 nm, positioned either at the center of the world or 1 $\mu$m away (center-to-center) along the vertices of a hexagonal polygon centered at the origin of the world (i.e., the edge-to-edge distance was 500 nm for consistency with [5]). The two hemispheres were separated from each other by a 20 nm high synaptic cleft. Four synapses (i.e., 4/7 = 57%) had 43% of their synaptic perimeter flanked by a 50 nm thick, 250 nm tall portion of an astrocyte, which we represented as a section of a cylindrical sheet located 20 nm away from the edge of the synapse. This organization is consistent with previous EM analysis of rodent hippocampal synapses [4, 5]. Each transporter molecule had a surface diffusion coefficient of $0.23 \cdot 10^{-8}$ $cm^2/s$ [2]. We used this model to estimate the glutamate concentration profile at all synapses. The glutamate transporters were modeled using a simplified kinetic scheme [7]. This scheme captures the known affinity and recovery time constant of glial glutamate transporters measured experimentally [9, 31, 54], and does not require an explicit simulation for binding/unbinding reactions of co-transported ions [55]. In this model, each transporter can interact with a diffusing glutamate molecule. All rates were adjusted for $Q_{10} = 3$, to approximate the transporter kinetics at 35˚C. In a set of simulations, we varied the surface density of the glutamate transporters between 0–20,000 $\mu m^{-2}$. CellBlender computes the surface area of the glial sheets, and populates them with a randomly distributed group of glutamate transporter molecules. Their number is calculated by CellBlender by multiplying the surface area and the chosen surface density value. At $t$ = 0, 2,000 glutamate molecules were released from a point source placed at the origin of the world, at the center of the synaptic cleft of the central synapse. Glutamate diffused away with a diffusion coefficient $D^* = 3.30 \cdot 10^{-6}$ cm$^2$/s within the synaptic cleft [56]. Outside the cleft the apparent diffusion coefficient was set to $D^* = 1.29 \cdot 10^{-6}$ cm$^2$/s, to account for a tortuosity value of $\lambda = 1.6$ [7]. Each simulation consisted of 100,000 iterations with a time step $t$ = 1 $\mu$s (therefore spanning a 100 ms time window), and was repeated 100 times. Glutamate hitting the surfaces of all objects in the simulation environment bounced back in the extracellular space (i.e., the surface of all other geometries in the simulation was reflective for glutamate). The glutamate concentration was monitored every $t$ = 10 $\mu$s in three regions: the volume above the PSD, the perisynaptic and extracellular volumes. The volume

above the PSD region was defined as the volume of a cylindrical portion of the synaptic cleft located above the PSD (i.e., a cylinder with radius $r = 125$ nm and height $h = 20$ nm). The PSD represented the central half of the apposition zone (i.e., a circle with radius $r = 125$ nm). The perisynaptic region represented the outer portion of the apposition zone surrounding the PSD (i.e., a cylindrical annulus with radius $r = 125$ nm and height $h = 20$ nm). The extracellular volume represented the volume of the world without including the synapses and glial sheets. We used these simulations to calculate the glutamate concentration in and out of the synaptic cleft. The glutamate concentration waveforms obtained from CellBlender were imported into ChanneLab 2.041213 (Synaptosoft) and used to calculate the open probability of AMPA and NMDA receptors. Given the limited surface density of these receptors, and the stochastic nature of their behavior, this approach allows to gain accurate predictions about their open probability without having to incur in prohibitively long simulation times [6, 32, 57, 58]. Therefore, it improves the computational efficiency of the simulations without compromising their accuracy. The kinetic rates for AMPA receptors were set in accordance to [59] and the kinetic rates for NMDA receptors were set in accordance to [60], which were both estimated at glutamatergic hippocampal synapses. In both cases, the reaction rates were adjusted by $Q_{10} = 3$, to approximate the receptor and transporter kinetics at 35˚C.

## Statistical analysis

Data are presented as mean±SD, unless otherwise stated. Statistical significance was determined by Student's paired or unpaired t-test, as appropriate (IgorPro 6.36). Differences were considered significant at p<0.05 (*p<0.05; **p<0.01; ***p<0.001). Pearson's correlation coefficient was calculated to compare Sholl distributions.

## Generation of 3D models of astrocytes

Our *in silico* rendition of astrocytes was based off our key features from the morphological analysis of biocytin-filled astrocytes processed with proExM and imaged with 2P-LSM, following conversion for a linear expansion factor of 2 [52]. A detailed description of how we distributed transporter trimers using different geometries is shown in S1–S5 Figs. Each model astrocyte consisted of a spherical soma soma and a tree of processes that branched off of it. The main features of reconstructed and model astrocytes are summarized in Table 3. The astrocyte branching tree was created recursively from a set of primary branches emerging from the soma. The model was developed using custom-made routines written in MATLAB2019 (Mathworks, Natick, MA). The recursive branching mechanism was based on the principle that new branches, with higher branching level, can be formed by each existing branch. The branching algorithm allowed us to control the number of branch levels, their length and diameter, and allowed for cell-to-cell branch variability. The algorithm incorporates a number of experimentally-derived constraints and features, which are described below.

**Astrocyte branch generation.** The number of primary branches departing from the soma was chosen randomly between $3 - 7$ for each model astrocyte, consistent with the number of primary branches measured through the morphological analysis of biocytin-filled astrocytes (Table 3). The primary branches emerged at random locations from the soma, and extended radially along straight lines. Each primary branch could generate up to $M = 15$ daughter (secondary) branches in a 3D space (at intervals that will be described below). The opening angles of the daughter branches, calculated with respect to the direction of the primary branch, were all set to $\pi/6$, and their rotation angles were chosen randomly within the range from $0 - 2\pi$. The operation was then repeated recursively for up to $N = 11$ branching levels, consistent with the maximum number of branching levels detected experimentally. At each branching point,

**Table 3. List of parameters describing astrocyte morphology *in vitro* and *in silico*.** Morphological measures were determined experimentally from 3D axial STEM tomography reconstructions ($\delta_n$) and from biocytin-filled neurons processed with proExM and imaged with 2P-LSM ($r_{soma}$, $R_{astro}$, $n_I$, $\chi_{0-9}$, $\lambda_0$). These measures were used to implement the *in silico*, model representations of astrocytes. Abbreviations: number of primary branches (num. branch I), distance (dist.), branching point (bpt), primary branch (I), secondary branch (II).

| Var | Description | Experiments | | | Model | | |
|---|---|---|---|---|---|---|---|
| | | mn (*μm*) | SD | n | mn (*μm*) | SD | n |
| $\delta_n$ | tip/leaf diameter | 0.25 (0.02–0.45) | 0.10 | 73 | 0.08 (0.02–0.19) | 0.04 | 24,556 |
| $r_{soma}$ | soma radius | 4.9 | 1.1 | 12 | 5 | 0 | 10 |
| $R_{astro}$ | cell radius | 112.2 | 23.6 | 10 | 100.9 | 32.5 | 10 |
| $n_I$ | num. branch I | 4.5 | 1.9 | 12 | 5.4 | 1.3 | 10 |
| $\chi_0$ | dist. soma – bpt 1 (I) | 6.5 | 4.1 | 41 | 7.2 | 4.2 | 54 |
| $\chi_1$ | dist. bpt 1 – 2 (I) | 4.4 | 3.4 | 31 | 5.9 | 4.2 | 54 |
| $\chi_2$ | dist. bpt 2 – 3 (I) | 4.8 | 1.9 | 14 | 5.8 | 4.6 | 54 |
| $\chi_3$ | dist. bpt 3 – 4 (I) | 3.4 | 2.7 | 19 | 6.4 | 3.9 | 54 |
| $\chi_4$ | dist. bpt 4 – 5 (I) | 2.4 | 2.4 | 10 | 5.8 | 4.3 | 54 |
| $\chi_5$ | dist. bpt 5 – 6 (I) | 9.6 | 12.1 | 2 | 6.5 | 4.1 | 54 |
| $\chi_6$ | dist. bpt 6 – 7 (I) | - | - | - | 6.0 | 3.7 | 54 |
| $\chi_7$ | dist. bpt 7 – 8 (I) | - | - | - | 6.0 | 4.3 | 54 |
| $\chi_8$ | dist. bpt 8 – 9 (I) | - | - | - | 5.7 | 5.2 | 51 |
| $\chi_9$ | dist. bpt 9 – 10 (I) | - | - | - | 4.3 | 2.9 | 22 |
| $\lambda_0$ | dist. base—bpt 1 (II) | 4.3 | 4.4 | 41 | 6.0 | 4.0 | 451 |

each parent branch could either generate a daughter branch and continue on, or terminate (see termination criteria below). The angle between the direction of the daughter branch and that of the parent branch increased from $\pi/6$ (as established for the secondary branches) by $\pi/80$ at each branching level. This was a simple linear model that best approximated the increasing trend in the size of the branching angles at each branching level, as observed in the reconstructed astrocytes. The segments connecting any two branching points were approximated as cylinders. An exception was made for the tips of the astrocyte terminal processes, which were modeled as hemispheres. The geometry at the tips and around branch points, as well as their consequences to the shape of the cell and transporter placement are further discussed in a separate section of this manuscript.

**Astrocyte branch length generation.** We generated the astrocyte branch segments (i.e., the portion of a branch between two consecutive branching points) in a way that would best fit our average empirical measures, while also accounting for the existence of cell-to-cell, in-cell and branch-to-branch variability. For each astrocyte, we first generated a random value $c_0$ within the range $0 - 25$ $\mu$m, to be used as a cell-wide baseline for segment lengths. In order to allow variability between primary branches (which exists in real astrocytes) we customized a specific segment length baseline for each primary branch, by randomly choosing a specific value of $c \le c_0$. All segments within the tree generated by a primary branch (i.e., the segments along the primary branch and all its daughter branches) were generated by adding a random term $\le 10$ $\mu$m to the baseline value $c$. This modeling scheme allowed us to best fit a collection of experimentally-derived measurements: the branch segments length ($\chi_0 - \chi_9$ and $\lambda_0$), the average branch length for each branching level, and the Sholl profile of the astrocytes.

**Astrocyte branch diameter generation.** The diameter of the cylinders representing the primary branches was set to 5 $\mu$m. To establish a rule to determine the diameter of the daughter branches, we considered previous attempts to solve this problem in neurons. In neurons, the input conductance of dendritic branches depends on the 3/2 power of the branch diameter [61]. When the constraint that the sum of the 3/2 power values of the daughter branch

diameters is equal to the 3/2 power of the parent branch diameter, the entire dendritic tree of the neuron can be mapped to an equivalent cylinder (provided also that all terminal branches end at the same electrotonic distance from the trunk). This rule is commonly referred to as the 3/2 rule [62, 63]. We asked whether this rule also applied to astrocytes. To do this, we measured the average diameter of the primary ($\delta_1 = 1.3 \pm 0.7 \, \mu$m, n = 46) and secondary branches ($\delta_2 = 0.8 \pm 0.3 \, \mu$m, n = 76) in a group of 10 astrocytes. We then calculated the values of $\delta_2$ that could be obtained once we set $\delta_1 = 1.3 \pm 0.7 \, \mu$m, using various values for $k$ in the relationship $D^{k/2} = d_1^{k/2} + d_2^{k/2}$, where $d_1 = d_2$. The most accurate prediction of the experimental data (i.e., $\delta_2 = 0.8 \pm 0.3 \, \mu$m) was obtained using the 3/2 rule. These findings suggested that the 3/2 rule holds true for both neurons and astrocytes. Based on this information, when creating daughter branches in our model, we determined the diameter $d_1$ of the parent branch after the branching point and the diameter $d_2$ of the daughter branch based on the diameter $D$ of the parent branch before the branching point, based on the 3/2 rule: $D^{3/2} = d_1^{3/2} + d_2^{3/2}$ [61–63]. To account for variability in the size of emerging branches, we set $d_1 = (f \cdot d_2)$, and used the 3/2 rule in the form: $D^{3/2} = (f \cdot d_2)^{3/2} + d_2^{3/2}$. Here, $d_2$ is the diameter of the daughter branch, and $(f \cdot d_2)$ is the diameter of the parent branch after the branching point. In our simulations, the bias $f$ was chosen randomly within the interval $1 - 1.5$.

**Astrocyte branch termination conditions.**   While all astrocytes were allowed to have up to $M = 15$ branching points along each primary branch, and up to $N = 11$ branching points in higher order branches, our algorithm also allowed us to terminate a branch at any branching point, based on their diameter. In each model astrocyte, the termination diameter $d_{max}$ was generated randomly within the interval $0.02 - 0.15 \, \mu$m for each primary branch (leading to an overall terminal diameter range comparable to that established based on 3D axial STEM tomography analysis of astrocytic processes [57]). The generation of daughter branches along a parent branch was also terminated if the diameter of any of the two emerging branch segments (as computed according to the 3/2 rule described above) was lower than $d_{max}$. The resulting range for tip diameters in model astrocytes was similar to that obtained from biocytin-filled astrocytes (Table 3).

**Astrocyte leaves.**   Small processes, called leaves, were added to all astrocyte branches. Each leaf was modeled as a cylinder of 0.7 $\mu$m length, randomly distributed every 3–8 $\mu$m along the length of each branch, and emerging perpendicularly from the main axis of each branch. The length and orientation of the leaves were taken to be consistent with those used in an existing study by Savtchenko et al. [64], in which leaves reconstructed using 60 nm thick serial EM sections were modeled as orthogonal to the parent branch, and with a length of 0.5–1.25 $\mu$m (the mean of which matches the length of our leaves). The linear density distribution of the leaves in our model was taken to be uniform, with a mean value that was calculated *post-hoc*, so that dependent measures (such as branch density per neuropil volume, leaf diameter and Sholl profile) were in agreement with our own empirical data. The diameter of each leaf was generated using the same 3/2 rule used for the generation of higher order processes, albeit with a larger bias $f$. Here, the bias $f$ was chosen in the interval 1.5–100, in a way that adapted to the thickness of the parent branch, but also incorporated a random aspect. Accordingly, $f = 0.5 + 3(1 + \rho)D$, where $\rho$ is a random number in the interval [0, 1], and $D$ is the parent branch diameter (the dependence on $D$ was introduced so that thicker branches would not generate thicker leaves). This way, leaf diameters in our model ranged between 0.02–0.19 $\mu$m (Table 3). Since the generation of each leaf with the 3/2 rule had only a minor impact on the branch diameter, the branch diameter was considered (for all computations) unchanged along the length of each branch, and only updated at the tip of the branch, according to the number of leaves on the segment. These leaves were not included in the Sholl analysis, as their size falls

below the resolution limit of the 2P-LSM used to detect the main morphological features of astrocytes after proExM.

**Morphology analysis of model astrocytes.** Each *in silico* generated astrocyte was designed to incorporate fixed terms (i.e., the somatic radius, the diameter of the primary branches, the maximum number of branch levels and the opening angle for each level) and random terms (i.e., branch length, diameter ratios are branch points, terminal diameter). This allowed our 3D models to produce a large variety of astrocyte sizes and shapes, consistent with experimental data. For each model astrocyte, we computed the cytoplasmic volume and total branch surface area, total branch length, average segment length $\lambda$ and $\chi$, average tip diameter, number of branches per level and average branch length per level. For each astrocyte, we computed the 3D Sholl profile, using 0.5 $\mu$m spaced spherical shells. We illustrate these 3D Sholl profiles of 10 randomly generated astrocytes, as well as their average, which agrees with the experimentally derived 3D Sholl profiles.

**Volume distribution measures.** Our model of the branching tree captures some important features of the 3D structure of astrocytes derived experimentally. For example, the core structure of our model (i.e., soma and branches larger than 1 $\mu$m) accounts for about 35±2% of the total astrocytic volume, while the smaller branches and leaves (the non-core structure) account for the rest of the 65±2% of the volume (n = 10). This is in general agreement with what was reported by [65] (i.e., $\sim$ 75%). In addition, the non-core structure in our model cells occupies 6.7±3.5% of the surrounding neuropil volume (n = 10), which is within the range reported by [66] (i.e., $\sim$ 4.0 − 8.9%). To compute this, we simulated the surrounding neuropil by covering the non-core structure with small cubes of side length of 2 $\mu$m, as follows. *First*, the whole 3D volume occupied by the astrocyte was partitioned into small cubes of 2 $\mu$m side length. From this partition, only the cubes intersecting non-core branches were retained and added to the volume of the simulated neuropil. We then computed the ratio between the actual volume of the non-core structure of the model, and the simulated neuropil, producing the estimate reported above. Using the same approach of covering the volume occupied by branches with cubes of side length 2 $\mu$m, we obtained a density for astrocytic processes in the *in silico* model of 4.3 ± 2.0 $\mu m^{-3}$, which is consistent with the density of 4.0 $\mu m^{-3}$ collected from 3D axial STEM tomography data.

## Glutamate transporter spatial distribution

We calculated the maximum number of transporters that could be placed in each model astrocyte, and the fraction of the membrane surface area they occupied. Our estimates were based off the crystal structure of glutamate transporter trimers of the prokaryotic homolog *Glt*$_{Ph}$, which can be approximated to a triangular prism, with a base of 8 nm side length and 6.5 nm height [67]. The analysis of glutamate transporter placement on the membrane of model astrocytes required us to analyze much simpler geometrical scenarios, in 2D and 3D. We first considered how the geometry of these triangular prisms, which resemble rectangles in the 2D space, can optimally fit that of a circle, a sphere (approximating the shape of the astrocyte soma) and a cylinder (approximating the shape of each astrocyte branch segment). The optimization constraints stemmed from the curvature of the membrane surface, which introduced some room between adjacent trimers, to avoid collision of the portions of these trimers protruding into the astrocyte cytoplasm or in the extracellular space (depending on the concavity/complexity of the branching points). Our optimal distribution was based on trimers fitting into spheres or cylinders that abide by this constraint, and maximize the space available on the membrane surface. While the theoretical trimer distribution that we propose is not always possible precisely in the form described (due to geometric limitations), our formulas can

nonetheless be used as general close estimates, especially for the soma/branch radius ranges in which the model operates. Each trimer replaces a curved portion of a sphere or cylinder. We computed analytically the area of this portion, as a function of the (sphere/cylinder) radius and of the trimer size. Last, we used these estimates to calculate the maximum surface density and surface area occupied by these molecules over the entire astrocyte, taking into account the potential effects introduced by angle constraints and transporter crowding around branching points. Our measurements took into account changes in the protrusion of transporters in the astrocyte cytoplasm, due to transporters moving across different conformational states.

**Distributing trimers on the surface of a sphere.**   To estimate the maximum number of transporter trimers that can be placed on the surface of a sphere of radius $r_1$, we computed the maximum triangular trimer bases that can cover, without colliding, a smaller sphere of radius $r_{-1}$ (accounting for the depth of cytoplasmic protrusion of the transporters $r_1 - r_{-1}$). We did this using the geometry of lunar triangles. A spherical lune is defined as the surface that lies between two great circles that intersect each other on the surface of a sphere. In our case, for a fixed triangle in the $w$-triangulation, one can consider the three planes defined by each pair of adjacent vertices, together with the center of the sphere. Each plane defines a great circle on the sphere, and each pair of great circles delimits a lune. We considered the spherical lunar triangle at the intersection of these three lunes with identical areas $A_{lune} = 4\alpha r_{-1}^2$, where $\alpha$ is the angle between the great circles defining the lune (in this case, identical for all three lunes). Notice that, for a $w$-triangulation of the sphere, the entire spherical surface can be written as a union of all lunar triangles defined by the triangulation, leading to **Eq (S11)**. The area $A^{\Delta_s}$ of the lunar triangle can be calculated using Girard's Theorem by expressing the area of the sphere as:

$$4\pi r_{-1}^2 = 3A_{lune} - 4A^{\Delta_s}$$

This gives us an expression for the area of the lunar equilateral triangle in terms of the sphere radius and the lunar angle.

$$A^{\Delta_s} = r_{-1}^2(3\alpha - \pi) \tag{2}$$

The angle $\alpha$ was computed using trigonometry in the spherical triangle. Call $\gamma = a = b = c$ the small arc defined by each of the triangle vertices on the corresponding great circle, and use the identity: $\cos a = \cos b \cos c + \sin b \sin c \cos \alpha$):

$$\cos \alpha = \frac{\tan(\gamma/2)}{\tan \gamma} = \frac{1 - \tan^2(\gamma/2)}{2} \tag{3}$$

Since $\sin\left(\frac{\gamma}{2}\right) = \frac{\omega}{2r_{-1}}$, the area of the lunar triangle corresponding to an $w$-triangulation of a sphere of radius $r_{-1}$ is:

$$A^{\Delta_s} = r_{-1}^2(3\alpha - \pi) = r_{-1}^2 \left[ 3 \arccos\left( \frac{2r_{-1}^2 - \omega^2}{4r_{-1}^2 - \omega^2} \right) - \pi \right] \tag{4}$$

**Computation of the sphere surface area occupied by transporters.**   The triangular base of a glutamate transporter trimer can be viewed in a reference $(x, y)$ coordinate plane as an equilateral triangle of side $\omega$ centered at the origin. We calculated the surface area $A^{\Delta}_{sphere}$ of the portion of the sphere of radius $r_1$ situated above this triangle. This represents the portion of the astrocyte surface area occupied by a transporter trimer of side length $w$. The computation of

the integral was carried out in polar coordinates, for convenience:

$$\frac{A^{\Delta}_{sphere}}{3} = \iint_D \sqrt{\left(\frac{\partial z}{\partial x}\right)^2 + \left(\frac{\partial z}{\partial y}\right)^2 + 1}\, dA = \iint_D \frac{r_1}{\sqrt{r_1^2 - x^2 - y^2}}\, dA$$

$$= \int_{\theta=-\pi/6}^{\theta=\pi/2} \int_{r=0}^{r=\frac{\omega}{2\sqrt{3}\sin(\theta + \pi/3)}} \frac{r_1}{\sqrt{r_1^2 - r^2}}\, r\, dr\, d\theta$$

$$= r_1^2 \int_{\theta=-\pi/6}^{\theta=\pi/2} 1 - \sqrt{1 - \frac{\omega^2}{12 r_1^2 \sin^2(\theta + \pi/3)}}\, d\theta$$

With the substitution $\tau = \theta + \pi/3$ and with the notation $\alpha = \dfrac{\omega}{2\sqrt{3}r_0}$, we obtained:

$$A^{\Delta}_{sphere} = 3r_1^2 \int_{\theta=\pi/6}^{\theta=5\pi/6} 1 - \sqrt{1 - \frac{\alpha^2}{\sin^2(\tau)}}\, d\tau \tag{5}$$

To derive an explicit dependence of $A^{\Delta}_{sphere}$ on parameters, we solved the remaining integral with the help of Integral Calculator (https://www.integral-calculator.com, accessed March 17, 2019):

$$F(\tau) = \int \sqrt{1 - \frac{\alpha^2}{\sin^2(\tau)}}\, d\tau$$

$$= \alpha \sin^{-1}\left(\frac{\alpha \cot \tau}{\sqrt{1 - \alpha^2}}\right) - \tan^{-1}\left(\frac{\cos \tau}{\sqrt{\sin^2 \tau - \alpha^2}}\right) \tag{6}$$

where $\alpha$ can be considered small enough so that $\alpha^2 < 1/2$ (which implies $\sin^2 \tau - \alpha^2 > 0$, since $\pi/6 \leq \tau \leq 5\pi/6$). The definite integral can then be calculated as:

$$\int_{\pi/6}^{5\pi/6} \sqrt{1 - \frac{\alpha^2}{\sin^2(\tau)}}\, d\tau = F(5\pi/6) - F(\pi/6) =$$

$$= 2\tan^{-1}\left(\frac{\sqrt{3}}{\sqrt{1 - 4\alpha^2}}\right) - 2\alpha \sin^{-1}\left(\frac{\sqrt{3}\alpha}{\sqrt{1 - \alpha^2}}\right) \tag{7}$$

$$= 2\tan^{-1}\left(\frac{3r_1}{\sqrt{3r_1^2 - \omega^2}}\right) - \frac{\omega}{\sqrt{3}r_1} \sin^{-1}\left(\frac{\sqrt{3}\omega}{12r_1^2 - \omega^2}\right)$$

In conclusion:

$$A^{\Delta}_{sphere} = 2\pi r_1^2 - 3r_1^2 \left[2\tan^{-1}\left(\frac{3r_1}{\sqrt{3r_1^2 - \omega^2}}\right) - \frac{\omega}{\sqrt{3}r_1} \sin^{-1}\left(\frac{\sqrt{3}\omega}{\sqrt{12r_1^2 - \omega^2}}\right)\right] \tag{8}$$

**Computation of the cylinder surface area occupied by transporters.** We called $A^{\Delta}_{cyl}(r_{cyl})$ the area that each trimer occupies on the surface of a cylinder of radius $r_{cyl}$. This can be calculated as the area of the cylindrical region lying on top of the domain $D$, enclosed by an equilateral triangle of side $\omega$ centered at the origin. Since the cylinder of radius $R$ is given by the equation $z = \sqrt{r_{cyl}^2 - x^2}$, the area of the portion of the cylinder lying above the triangle can be

expressed as the double iterated integral:

$$
\begin{aligned}
A_{cyl}^{\Delta}(r_{cyl}) &= \iint_{D} \sqrt{\left(\frac{\partial z}{\partial x}\right)^2 + \left(\frac{\partial z}{\partial y}\right)^2 + 1}\, dA = 2 \int_0^{\omega/2} \int_{-\omega/2\sqrt{3}}^{\omega/\sqrt{3}-\sqrt{3}x} \frac{r_{cyl}}{\sqrt{r_{cyl}^2 - x^2}}\, dy\, dx \\
&= \sqrt{3}\,\omega r_{cyl} \sin^{-1}\left(\frac{\omega}{2r_{cyl}}\right) + \sqrt{3}\,r_{cyl}\sqrt{4r_{cyl}^2 - \omega^2} - 2\sqrt{3}\,r_{cyl}^2
\end{aligned}
\tag{9}
$$

Additional information on glutamate transporter distribution in tips/leaves and branching points can be found in S5 Fig.

## Results

### Quantification of glutamate transporters in mouse hippocampal astrocytes

We quantified GLAST and GLT-1 protein levels in membrane protein extracts of the mouse hippocampus using dot blotting experiments. As a first step, we dissected the hippocampi from both brain hemispheres from mice aged 2 weeks to 21 months. Hippocampi from 7–8 week old rats were used to validate our approach, as this is the same age range of rats used to derive previous estimates of GLAST and GLT-1 surface density [11]. We extracted all proteins from the hippocampi (i.e. cytoplasmic and membrane-bound), and determined their concentration using a Bradford assay. Next, the hippocampal membrane protein extracts from the mouse hippocampus in each age cohort were dotted alongside known amounts of purified, recombinant proteins of the human homologs of GLAST and GLT-1 (Fig 1). The intensities of the dots was measured as raw integrated density (RID) using image analysis software. The RID of the purified proteins was used to generate a standard curve, which was used to convert the RID value of each sample dot to weights and then moles of GLAST or GLT-1 (Fig 1A–1D).

We used the dot blot results to estimate the surface density of each glutamate transporter in hippocampal astrocytes. Like [11], we assumed that: *(i)* the mouse and rat hippocampus contain $\sim 97.8$ g of protein per Kg of tissue [68]; *(ii)* the rodent hippocampus has a density of 1.05 g/cm$^3$ [69]; *(iii)* GLAST and GLT-1 are mostly expressed in astrocytes (only $\sim 5-10\%$ of these molecules is expressed in neurons [11, 19]); *(iv)* GLAST and GLT-1 are mostly expressed in the plasma membrane (their level of expression in association with intracellular membranes is small [11, 16, 70]); *(v)* the molecular weights of GLAST and GLT-1 are 59,619 Da and 62,027 Da in mice, and 54,374 Da and 60,914 Da in rats, respectively [71, 72].

The estimates reported in Fig 2A indicate that GLAST and GLT-1, together, contribute at most $\sim 6\%$ of all hippocampal proteins in 6 month old mice. This value is $\sim 4.5\%$ in 7–8 week old mice and $\sim 3.5\%$ in 7–8 week old rats, meaning $\sim 2.2-2.8\%$ times higher than those previously reported for 7–8 week old rats by [11]. The reasons for this small difference might be due to different protein extraction procedures or to existing differences between rat strains, which are beyond the scope of this work. According to our experiments, GLAST and GLT-1, together, comprise up to 15% of all hippocampal membrane proteins in 6 month old mice, with varying levels depending on mouse age Fig 2B. In the rat visual cortex, estimates from 2D EM images suggest that the astrocyte membrane density is 1.3–1.6 $\mu m^2/\mu m^3$ [12]. We tested whether these measures hold true for the mouse hippocampus by using a more direct approach, capable of detecting astrocyte membranes in 3D. To do this, we identified astrocytic processes in *stratum radiatum* of the mouse hippocampus using 3D axial STEM tomography, a technique that offers a spatial resolution of only a few nm [49, 50, 73] and that allowed us to analyze a tissue block of $\sim 18\ \mu m^3$ volume (Fig 3A). We traced the contours of astrocytic processes manually (Fig 3B), to generate 3D reconstructions (Fig 3C) [52, 57]. This allowed us to

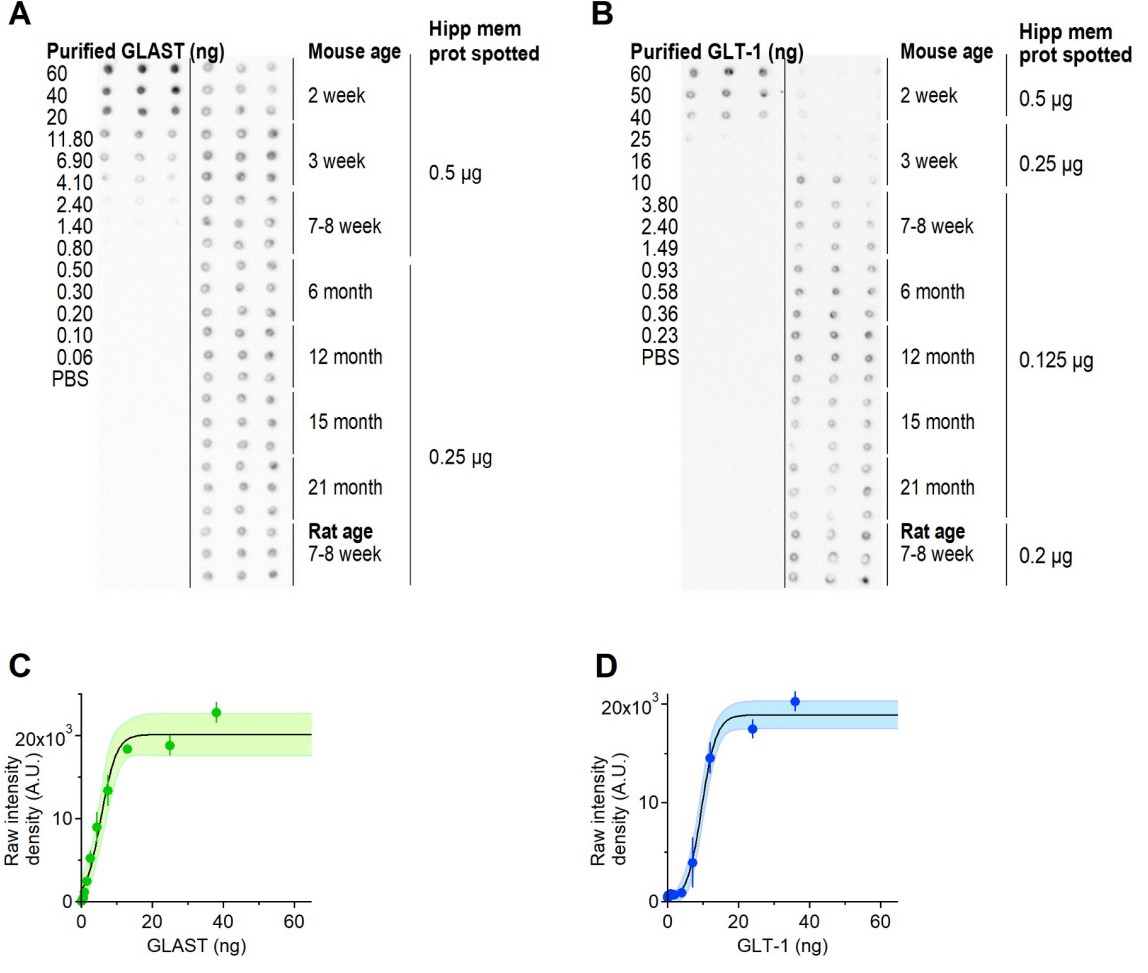

**Fig 1. Dot blot analysis of GLAST and GLT-1 expression in the mouse hippocampus. A**. Dot blots of recombinant and native GLAST in hippocampal membrane protein extracts from mice aged from 2 weeks to 21 months, and rats aged 7–8 weeks. **B**. As in A, for GLT-1. **C**. Sigmoidal fit of dot intensity for different amounts of recombinant GLAST. Each dot represents the mean±SD (n = 3). **D**. As in C, for GLT-1.

estimate the total surface area of small astrocytic processes contained in this tissue block, as well as their size and density ($\delta_n$ in Table 3).

The astrocyte membrane density in the reconstructed block was 3.15 $\mu$m$^2$/$\mu$m$^3$, which is more than two times larger than previous values obtained from extrapolations made using 2D EM micrographs in layers I-IV of the visual cortex of Long Evans male rats aged P23–25 [12], and previously used to estimate the astrocyte plasma membrane density in the rat hippocampus [11]. The density of small astrocytic processes was ∼ 4 $\mu m^{-3}$. Comparative analysis shows that the proportion of neuronal versus non-neuronal cells in mice and rats are very similar, and the shape and size of non-neuronal cells is very similar in the neocortex, cerebellum and the rest of the brain of these rodent species [74]. Therefore, we used our estimated value of astrocytic membrane density (i.e., 3.15 $\mu$m$^2$/$\mu$m$^3$) to calculate the surface density of glutamate transporters in our own tissue samples, from mice and rats. We assumed that there is no major change in the surface area of astrocytes in the hippocampus of mice aged 2 weeks—21 months. The amounts of transporter proteins as percent of total tissue proteins and the tissue concentrations of glutamate transporters are shown in Fig 2C–2E and in Table 4, respectively.

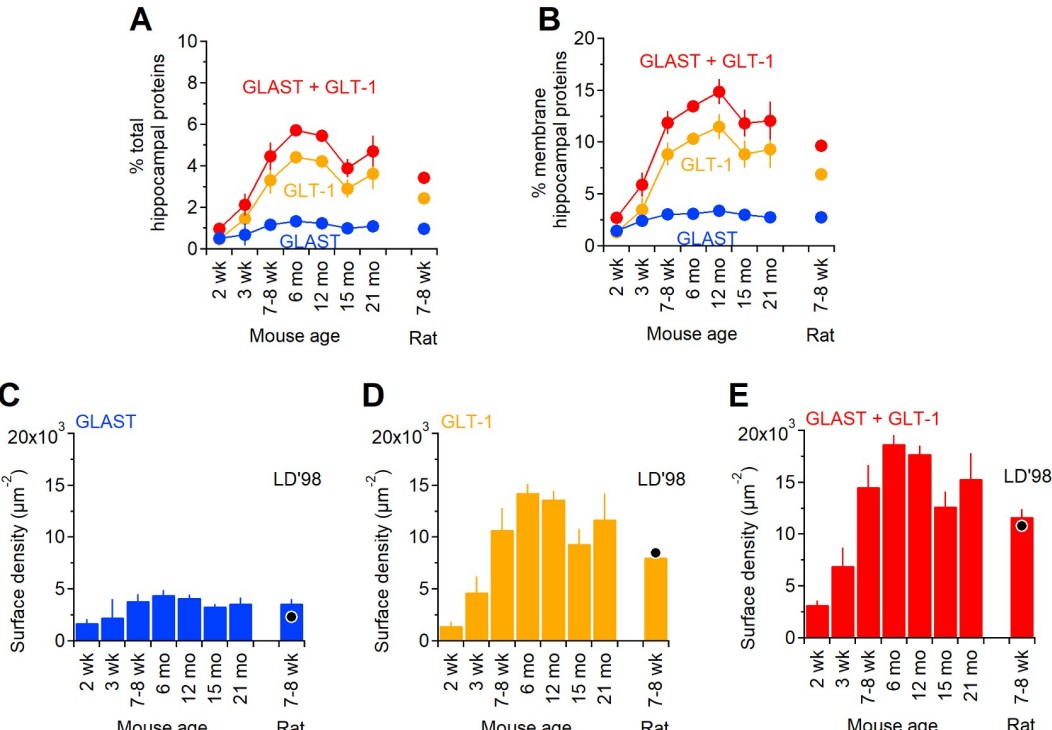

**Fig 2. Transporter proteins as percent of total and membrane-bound hippocampal proteins.** The amount of each glutamate transporter in mouse hippocampal membranes was determined by quantitative dot blot as described in the Materials and methods section. **A**. The percentage of total hippocampal proteins (i.e., cytoplasmic and membrane-bound) represented by each transporter was calculated from n = 3 samples, assuming that there are 97.8 g of protein per Kg of tissue, for GLAST (*blue*) and GLT-1 (*orange*). The sum of the data collected for GLAST and GLT-1 is shown in red. **B**. Percentage of hippocampal membrane proteins represented by GLAST (*blue*) and GLT-1 (*orange*). The concentration of total proteins in the hippocampal membrane fraction was determined by Bradford assay. The amount of transporter detected in each dot was divided by the total amount of hippocampal membrane protein spotted. **C**. Estimates of GLAST surface density expression in hippocampal astrocytes of mice aged 2 weeks—21 months and in 7–8 week old rats, based on the parameters listed in Table 4. The black dot represents the results obtained in previous estimates from 7–8 week old rats [11]. **D**. As in C, for GLT-1. **E**. Estimates of surface density for both GLAST and GLT-1. For consistency with previous work [11, 16], a correction for 5–10% neuronal GLT-1 was not applied to our surface density estimates. Data represent mean±SD.

The expression of GLT-1 increases between 2 weeks and 6 months, and then starts declining (note that there is no statistical significance between the data collected at 15 months and 21 months; p = 0.22). A similar trend was detected for GLAST and GLT-1, although the overall expression of GLAST is much lower than that of GLT-1. The values obtained in 7–8 week old mice are similar to those obtained in 7–8 week old rats, and consistent with those previously obtained in 7–8 week old rats by [11]. This suggests that our estimates provide a good approximation of glutamate transporter surface density in both rats and mice.

## An evaluation of glutamate transporter surface density based on their heterogeneous distribution along the astrocyte plasma membrane

A number of studies suggest that glutamate transporters are non-uniformly distributed along the plasma membrane of astrocytes, with a rapid recycling rate from intracellular compartments and a range of surface diffusion rates [2, 2, 3, 13, 16, 40, 42, 45]. Their expression is often described as punctate, with regions where transporters are enriched and display lower lateral mobility, which are confined to fine terminal branch tips and small lateral protrusions

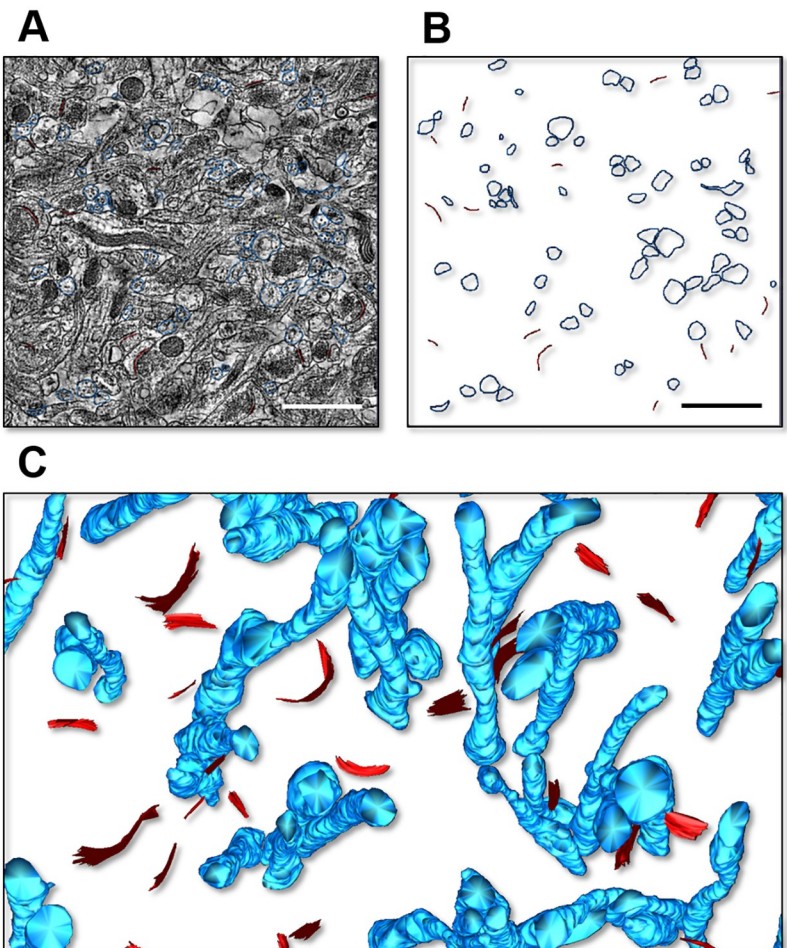

**Fig 3. 3D Axial STEM tomography analysis of astrocytic processes in the mouse hippocampus. A**. Example of a single plane of of the reconstructed block of tissue, with contours of astrocytic processes (*blue*) and post-synaptic density (PSD; *red*). Scale bar: 1 $\mu$m. **B**. As in B, without the raw EM data. **C**. 3D representation of the astrocytic processes and PSD regions in a portion of the reconstructed tissue block.

(leaves) present on all astrocytic processes [2, 13, 16, 40, 42]. Different glutamate transporter variants have been suggested to exhibit marked differences in their sub-cellular localization [48]. For example, GLAST tends to be excluded from astrocytes fine processes (i.e., branch tips and leaves), where its expression is 0.6-fold of that at the astrocyte soma [48]. Here, the branch tips represent the terminal ending of an astrocyte branch, whereas the leaves represent the thin lateral protrusions that depart from each branch. The expression ratio of GLAST between the astrocyte tips/leaves:shafts:stems is 0.8:0.6:1. Here, the stems represent the GFAP$^+$ primary and secondary branches, whereas the shafts represent all other regions of the astrocyte plasma membrane except the soma, branch tips and leaves [75]. This means that the tips/leaves:shafts: stems:soma expression ratio with respect to the soma is 0.6:0.5:0.8:1. In contrast, GLT-1 is primarily localized to the plasma membrane, with a 2.3-fold higher expression in fine processes compared to the soma [48]. For GLT-1, the expression ratio at the astrocyte tips/leaves:shafts: stems is 3.2:2:1 [48]. Therefore, the tips/leaves:shafts:stems:soma ratio with respect to the soma is 2.3:1.4:0.7:1. Although the astrocyte tips/leaves are in contact with a subset of excitatory synapses [4], this does not mean that other portions of the astrocyte plasma membrane cannot

**Table 4. Tissue concentrations of glutamate transporters.** The number of molecules of GLAST or GLT-1 per volume of hippocampal tissue and per cell membrane area were calculated from the dot blot experiment using the following assumptions: the mouse hippocampus contains 97.8 g of protein per Kg of tissue [68], has a density of 1.05 g/$cm^3$ [69], and that the molecular weigh of GLAST and GLT-1 are 59,619 and 62,027 Da, respectively (mean±SD; $n$ = 3). The astrocyte plasma membrane density was set to 3.15 $\mu m^2/\mu m^3$ for mice (according to measures obtained from our 3D axial STEM tomography data) and rats (see Results section for details).

| | age | mg transp / g hipp tissue | $\mu M$ | molec / $\mu m^3$ | $\mu m^2$ / $\mu m^3$ | molec / $\mu m^2$ |
|---|---|---|---|---|---|---|
| GLAST | 2 wk | 0.51 ± 0.10 | 9 ± 2 | 5,399 ± 1,026 | 3.15 | 1,714 ± 326 |
| | 3 wk | 0.67 ± 0.52 | 12 ± 9 | 7,147 ± 5,482 | 3.15 | 2,269 ± 1,740 |
| | 7–8 wk | 1.15 ± 0.19 | 20 ± 3 | 12,190 ± 1,972 | 3.15 | 3,870 ± 626 |
| | 6 mo | 1.32 ± 0.13 | 23 ± 2 | 13,980 ± 1,365 | 3.15 | 4,438 ± 433 |
| | 12 mo | 1.23 ± 0.09 | 22 ± 2 | 13,033 ± 976 | 3.15 | 4,138 ± 310 |
| | 15 mo | 0.98 ± 0.05 | 17 ± 1 | 10,409 ± 559 | 3.15 | 3,304 ± 177 |
| | 21 mo | 1.08 ± 0.15 | 19 ± 3 | 11,430 ± 1,618 | 3.15 | 3,628 ± 514 |
| | 7–8 wk rat | 0.97 ± 0.11 | 19 ± 2 | 11,333 ± 1,245 | 3.15 | 3,598 ± 395 |
| GLT-1 | 2 wk | 0.45 ± 0.12 | 8 ± 2 | 4,596 ± 1,264 | 3.15 | 1,459 ± 401 |
| | 3 wk | 1.45 ± 0.46 | 25 ± 8 | 14,783 ± 4,687 | 3.15 | 4,693 ± 1,488 |
| | 7–8 wk | 3.31 ± 0.64 | 56 ± 11 | 33,718 ± 6,564 | 3.15 | 10,704 ± 2,084 |
| | 6 mo | 4.40 ± 0.25 | 75 ± 4 | 44,880 ± 2,558 | 3.15 | 14,248 ± 812 |
| | 12 mo | 4.21 ± 0.25 | 71 ± 4 | 42,890 ± 2,502 | 3.15 | 13,616 ± 794 |
| | 15 mo | 2.89 ± 0.43 | 49 ± 7 | 29,473 ± 4,356 | 3.15 | 9,357 ± 1,383 |
| | 21 mo | 3.62 ± 0.75 | 61 ± 13 | 36,894 ± 7,671 | 3.15 | 11,712 ± 2,435 |
| | 7–8 wk rat | 2.44 ± 0.22 | 42 ± 4 | 25,344 ± 2,309 | 3.15 | 8,046 ± 733 |
| GLAST+GLT-1 | 2 wk | 0.96 ± 0.12 | 17 ± 2 | 9,996 ± 1,264 | 3.15 | 3,173 ± 401 |
| | 3 wk | 2.12 ± 0.52 | 36 ± 9 | 21,930 ± 5,482 | 3.15 | 6,962 ± 1,740 |
| | 7–8 wk | 4.46 ± 0.64 | 76 ± 11 | 45,909 ± 6,564 | 3.15 | 14,574 ± 2,084 |
| | 6 mo | 5.72 ± 0.25 | 98 ± 4 | 58,860 ± 2,558 | 3.15 | 18,686 ± 812 |
| | 12 mo | 5.44 ± 0.25 | 93 ± 4 | 55,924 ± 2,502 | 3.15 | 17,754 ± 794 |
| | 15 mo | 3.87 ± 0.43 | 66 ± 7 | 39,882 ± 4,356 | 3.15 | 12,661 ± 1,383 |
| | 21 mo | 4.70 ± 0.75 | 80 ± 13 | 48,323 ± 7,671 | 3.15 | 15,341 ± 2,435 |
| | 7–8 wk rat | 3.42 ± 0.22 | 61 ± 4 | 36,678 ± 2,309 | 3.15 | 11,644 ± 733 |

also form contacts with them. This makes room for an interesting consideration: that the average surface density measures for GLAST and GLT-1 derived from our dot blot experiments (Table 4) can be refined based on the known variations in the expression of each transporter variant across different sub-cellular compartments. To account for this, we first need to determine which proportion of the entire astrocyte plasma membrane is represented by the tips/leaves, shafts, stems and soma. This can be done using a modeling approach capable of capturing macroscopic features of the astrocyte morphology (captured experimentally using proExM and 2P-LSM) as well as microscopic ones (captured experimentally using 3D axial STEM tomography).

We first analyzed the morphological properties of biocytin-filled mouse hippocampal astrocytes using proExM, 2P-LSM imaging and then scaled them to their real size using a correction factor for linear expansion (see Materials and methods). From the 3D reconstructions of these astrocytes, we estimated the volume occupied by the whole cell and all its processes (astrocyte cytoplasm), the volume of the neuropil occupied by the astrocyte (astrocyte boundary), and the volume occupied by the astrocyte branches (Fig 4A). The branching levels were defined according to the color code shown in Fig 4C. The average volume of the neuropil occupied by an astrocyte was $5.9 \cdot 10^5 \pm 2.1 \cdot 10^5$ $\mu m^3$, whereas the average astrocyte cytoplasmic volume was $2.2 \cdot 10^5 \pm 1.2 \cdot 10^5$ $\mu m^3$ (n = 10; mean±SD). The volume occupied by the branch approximation of the astrocyte, which provided a lower bound for the cytoplasmic volume, was 8.2 ·

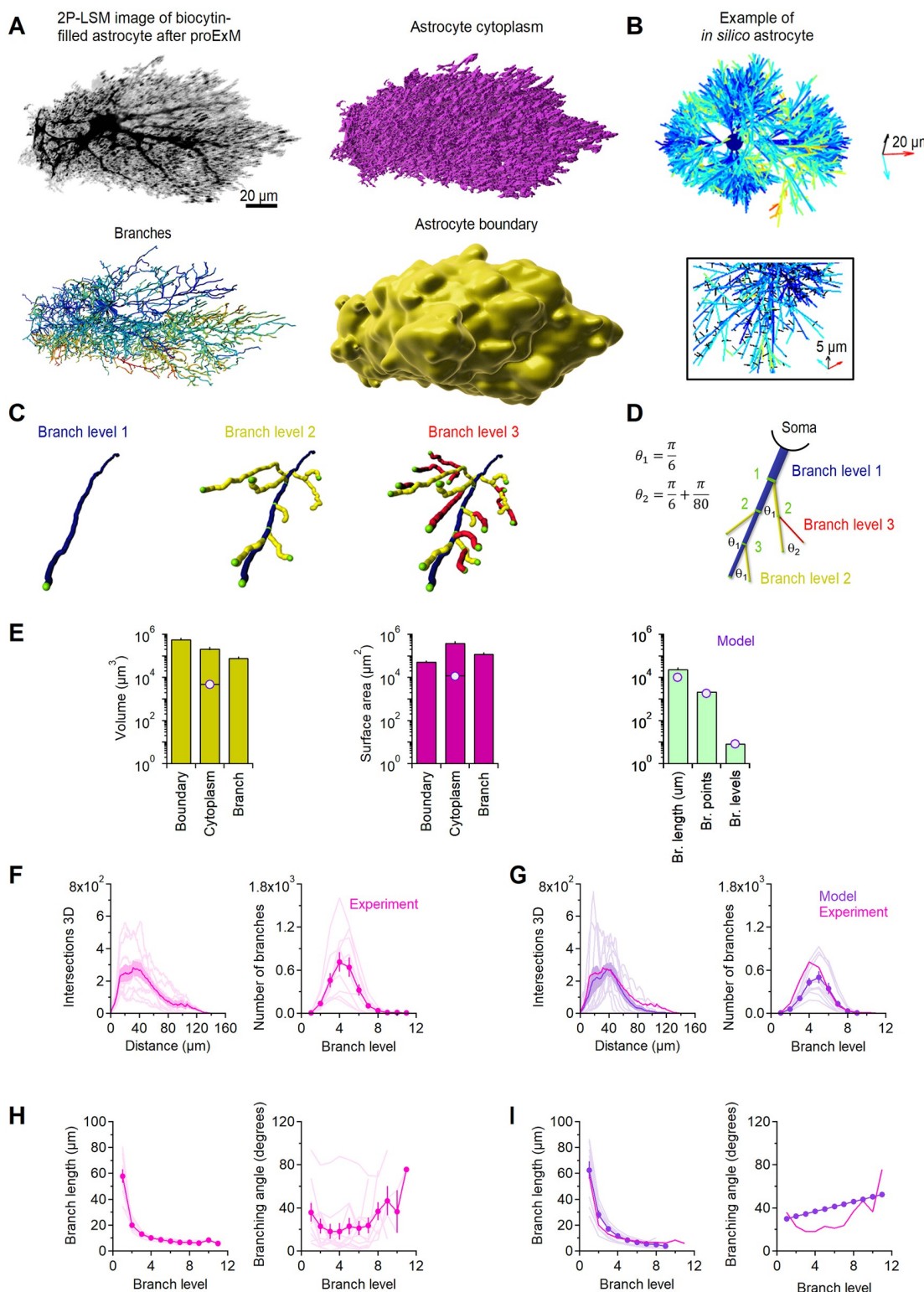

**Fig 4. Astrocyte morphology in *stratum radiatum* of the mouse hippocampus. A**. *Top left*, Maximum intensity projection of a 2P-LSM z-stack of a biocytin-filled astrocyte in a mouse hippocampal *stratum radiatum*. *Top right*, Image analysis of astrocytic branches. Higher branching levels are denoted by warmer colors. *Bottom left*, Image analysis of astrocyte cytoplasm. *Bottom right*, Volume of the hippocampal neuropil occupied by the astrocyte. **B**. *Top*, *In silico* representation of an astrocyte generated through our MATLAB model. The color scheme for each branch level is the same used for the reconstructions of biocytin-filled astrocytes

shown in panel A. The scale bar is 20 μm in the x (cyan), y (red) and z axis (black). *Bottom*, close-up view of a model astrocyte, with leaves color coded in black. **C**. Color-coded representation of branching levels of progressively higher order. **D**. As in C, for the *in silico* data. **E**. Summary analysis of volume (*left*) and surface area analysis (*middle*) of the cell boundary, cytoplasm and branches (volume, ***p = $2.7 \cdot 10^{-4}$; surface area, ***p = $1.6 \cdot 10^{-4}$). The purple dots represent values of volume and surface area obtained in the model without including leaves. The horizontal orange line represent values of surface area and volume obtained by including estimates obtained for leaves. The histogram on the right provide a quantitative analysis of branch length, number of branch points and branch levels across all analyzed astrocytes (length, **p = $4.2 \cdot 10^{-3}$; points, p = 0.49; levels, p = 0.65). **F**. Sholl analysis (*left*) and summary of branch count analysis for different branch levels (*right*). **G**. As in F, for the *in silico* data (intersections 3D, Pearson's correlation coefficient r = 0.98; number of branches, p = 0.77). **H**. Summary analysis of branch length (*left*) and branching angle (*right*) for each branching level. **I**. As in G, for the *in silico* data (branch length, p = 0.73; branch angle, p = 0.14). Data represent mean±SEM.

$10^4 \pm 3.0 \cdot 10^4 \ \mu m^3$ (n = 10; Fig 4E, *left*). These values are consistent with analogous measures obtained by [65]. The astrocyte surface area was $4.0 \cdot 10^5 \pm 2.0 \cdot 10^5 \ \mu m^2$ (n = 10; Fig 4E, *center*). On average, in each astrocyte, the cumulative length of all branches was $2.4 \cdot 10^4 \pm 1.2 \cdot 10^4 \ \mu$m, with $2.2 \cdot 10^3 \pm 1.3 \cdot 10^3$ branching points and $8.5 \pm 1.2$ branch levels (Fig 4E, *right*).

We used these measures as our reference to build the backbone of an *in silico* model of astrocytes (Fig 4B). While some aspects of our geometric model are compatible and agree with existing work (e.g., [64]), our model is unique in other aspects that are driven by our own experimental data and the specific aim of understanding cell-compartment specific differences in the distribution of glutamate transporters. Even though a full, point-by-point comparison with other existing reconstructive models is not possible, we will include, when applicable, discussion points around the overlap with other studies. In this model, the soma was approximated as a sphere and each process was approximated as a cylinder with rounded tips at their endings. At each branching point, the parent branch generated a daughter branch with a diameter that obeyed the 3/2 rule [61]. Each astrocyte branch was populated with leaves, represented as fine lateral cylindrical protrusions with rounded tips at their endings. At each branching point, the daughter branch had a smaller mean diameter and higher branch level than the parent branch. By contrast, the parent branch became thinner and maintained the same branch level (Fig 4D). As shown in our 3D axial STEM tomography data (Fig 3, Table 3), the size of the small processes that emerge from the main astrocyte branches (i.e., the leaves) has an average diameter of 250 nm, with a range of 20–450 nm. These small processes are present throughout the astrocyte surface and give these cells a spongiform appearance [76]. To account for the presence of leaves in our model, we used an approach similar to that of [64]. Accordingly, leaves were incorporated in our model as a multitude of short (i.e., 0.7 μm) and thin cylindrical extensions. Tips and leaves had a mean diameter of 0.08 ± 0.04 μm, consistent with the experimental values of $\delta_n$ reported in Table 3. Leaves emerged orthogonally with respect to the main axis of the parent branch at random intervals, so that their density in the neuropil was similar to that detected in our 3D axial STEM tomography measures (experiments: $4.0 \ \mu m^{-3}$; model: $4.3 \pm 2.0 \ \mu m^{-3}$). The average volume of the *in silico* astrocytes was $4.6 \cdot 10^3 \pm 0.8 \cdot 10^3 \ \mu m^3$ (***$p = 2.3 \cdot 10^{-4}$), with a surface area of $1.1 \cdot 10^4 \pm 2.9 \cdot 10^4 \ \mu m^2$ (***$p = 1.4 \cdot 10^{-4}$). The average branch length ($9.7 \cdot 10^3 \pm 2.4 \cdot 10^3 \ \mu m$; **$p = 1.3 \cdot 10^{-3}$), number of branching points ($1.9 \cdot 10^3 \pm 0.9 \cdot 10^3$; $p = 0.56$) and branching levels ($8.5 \pm 0.5$; $p = 1.00$) were in good agreement with the measures collected from biocytin-filled astrocytes reconstructed experimentally (Fig 4E, *right*). The model also provided a reasonably accurate description of the Sholl profile (Fig 4F–4G, *left*; ***$p = 2.1 \cdot 10^{-13}$), number of branches (Fig 4F–4G, *right*; $p = 0.64$), branch length (Fig 4H–4I, *left*; $p = 0.77$) and branching angle for each branch level of the astrocyte (Fig 4H–4I, *right*; $p = 0.14$). The surface to volume ratio of model astrocytes was $2.4 \pm 0.3 \ \mu m^{-1}$ without including leaves, and $2.6 \pm 0.2 \ \mu m^{-1}$ with leaves (n = 10). These values are consistent with those reported in previous work by [77] using confocal

morphometry ($\sim 2.64\ \mu m^{-1}$), and by [78] in the cat cortex ($\sim 3.3\ \mu m^{-1}$). The discrepancy between these values and much higher ones reported by [79–81] ($13.0-26.2\ \mu m^{-1}$) has been previously suggested to be due to cell swelling during the fixation procedure in EM measurements, as well as to the differences in image processing and resolution between confocal microscopy and EM [77]. In our model, the tips/leaves, together with the astrocyte shafts, are analogous to what [65] defined as the non-core regions. In our simulations, the sum of the volume of shafts and tips/leaves (i.e. the non-core regions) is $\sim 65\%$ of the entire astrocyte volume. This is consistent with the estimates of [65], where these non-core regions occupy $\sim 75\%$ of the entire astrocyte volume. In our astrocyte models, we also calculated the tissue volume fraction occupied by fine processes, as well as the density of processes in the neuropil. For consistency with empirically-based estimated of analogous measures, we simulated the neuropil of a model astrocyte as a minimal cover that can be obtained with a 3D partition of small cubes of 2 $\mu$m side length, as described in further details in the Materials and methods and in S6 Fig. With this computation method, the simulated tissue volume fraction, was 6.7 ± 3.5% (n = 10), which is consistent with the values of 4.0–8.9% reported by [66]. In addition, the branching pattern of our model astrocyte is such that there are, on average, 4.3 ± 2.0 $\mu m^{-3}$ processes in the neuropil covered by them, consistent with our 3D axial STEM tomography data showing that there are 4.0 $\mu m^{-3}$ processes in our sample tissue. Together, these findings indicate that our model is in general agreement not only with our experimental data and with data collected by other groups.

From the model astrocytes, we calculated that the astrocyte tips and leaves, shafts, stems and soma represented 1.8% ± 0.6%, 62.5% ± 6.2%, 34.1% ± 5.8% and 1.6% ± 0.6% of the total astrocyte plasma membrane, respectively (n = 10; mean±SD). Therefore, $\sim 98\%$ of the astrocyte membrane has a surface density for glutamate transporters that has higher GLAST and lower GLT-1 levels than those present at the tips/leaves of astrocytes (Fig 5E). The results, shown in Fig 5, indicate how different the concentration of transporters is in the tips/leaves, shafts, stems and soma. This suggests that the identity of the astrocyte sub-cellular compartment in contact with a synapses, not only the extent of its synaptic coverage, is a key determinant of glutamate spillover and extrasynaptic receptor activation at a given synapse. Synapses with the same levels of astrocyte coverage can experience various levels of spillover, where the maintenance of the specificity of synaptic transmission is dependent on which sub-cellular compartment of a given astrocyte reaches the synapse.

## Theoretical estimates of relative surface density and effective surface fraction occupied by glutamate transporters

There are geometrical constrains that prevent glutamate transporters from occupying specific domains of the plasma membrane due to crowding effects (S5 Fig). This is due to the fact that glutamate transporters are sizable molecules, and steric hindrance limits their expression at tight insertion points (i.e., the branching points) and in structures with convex curvature (i.e., branch tips and leaves; S5 Fig). As an example, the extracellular portions of glutamate transporters can collide with each other an branching points, whereas their cytoplasmic portions can collide with each other at the tips. This means that very small, restricted domains of the astrocyte membrane can lack glutamate transporters. These regions can be wider or narrower depending on the insertion depth in the lipid bilayer, which changes depending on the conformation state of the transporters. Accordingly, glutamate transporters protrude towards the cytoplasm when they are fully bound to glutamate (Fig 5F) [82]. For this reason, we asked the following two questions: *(ii)* how does the experimentally-derived surface density of glutamate transporters compare with the maximum one that can be achieved theoretically, given the

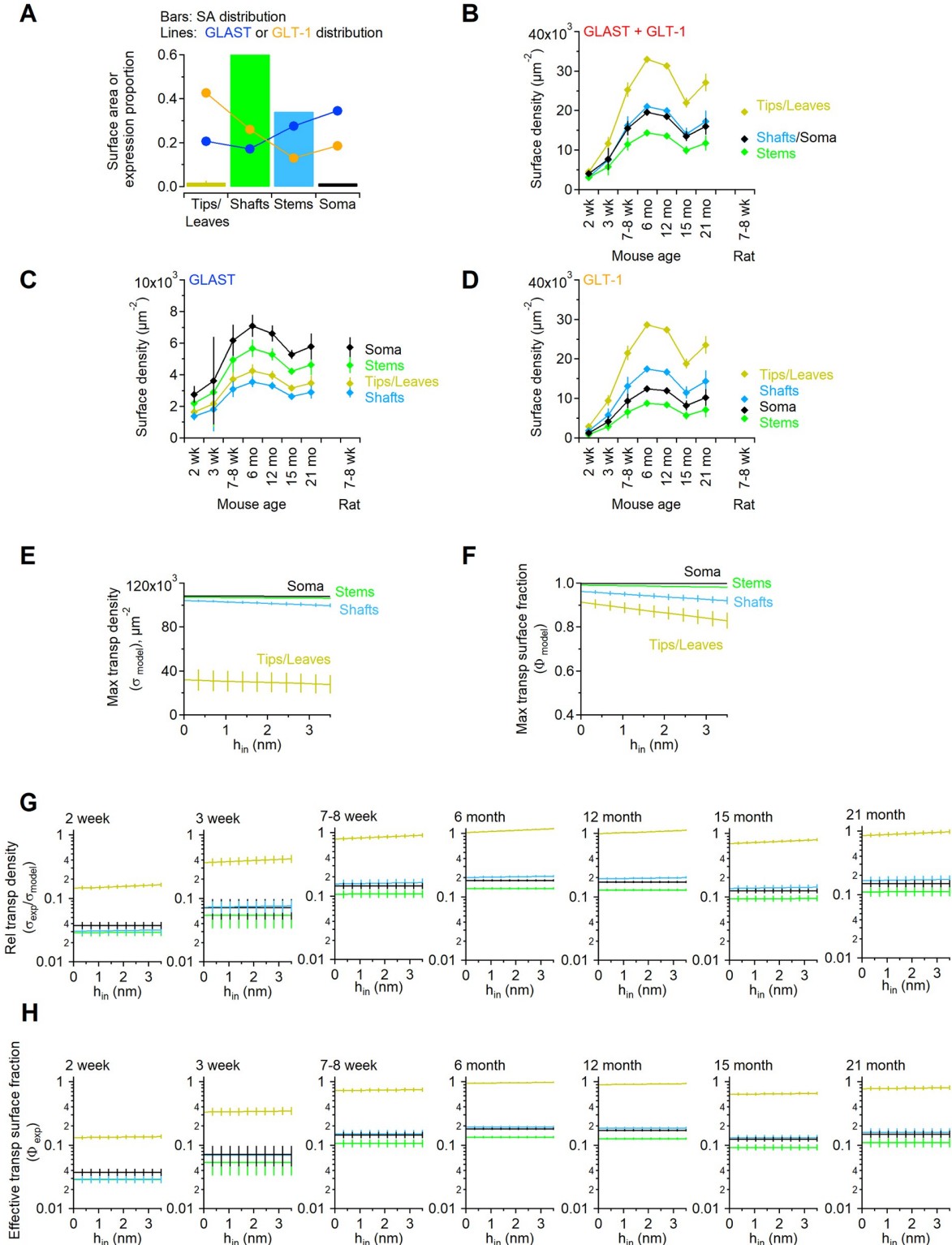

**Fig 5. Spatial gradient of glutamate transporter expression in different sub-cellular compartments. A**. The color-coded bar graphs represent the surface area distribution in four different astrocyte sub-cellular compartments. The tips and leaves represent the hemispheric endings of the terminal branches, and the thin protrusions that depart from all astrocyte branches, respectively. The stems represent the sum of primary and secondary branches. The soma represents the surface area of a spherical representation of the soma following subtraction of the surface area of the sphere underlying the base of the primary branches. The shafts represent all other compartments that

are not part of the ones described above. The blue line and filled circles described the relative distribution of GLAST in different sub-cellular compartments [48, 75]. Analogous measures for the distribution of GLT-1 are shown using the orange line and filled circles. For both GLAST and GLT-1 we scaled these proportions from [48, 75] so that their sum was 1. **B**. Surface density of glial glutamate transporters in different sub-cellular compartment of hippocampal astrocytes for mice aged 2 week—21 months and 7–8 week old rats. **C, D**. The data were calculated using the surface density values from Table 4, using distribution ratios in the tips/leaves:shafts:stems:soma of 0.6:0.5:0.8:1 and 2.3:1.4:0.7:1 for GLAST and GLT-1, respectively ($n = 3$). The weighted average of the data relative to each compartment corresponds to values shown in Table 4. Data represent mean±SD. **E**. Measures of the maximal surface density of glutamate transporters that can be achieved in the astrocyte tips/leaves, stems, shafts and soma. **F**. Maximal surface fraction that can be occupied by glutamate transporters in the astrocyte tips/leaves, stems, shafts and soma. **G**. Relative transporter surface density, measured as the ratio between the experimentally measured transporter density, distributed in each compartment (Fig 5), and the maximal one obtained from the model shown, for 2 week to 21 month old mice and for different insertion depths of glutamate transporters through the plasma membrane. **H**. Effective transporter surface fraction, calculated as the ratio between the surface area occupied by transporters and the maximum area that they can occupy in each compartment, for different insertion depths. In these graphs, the values of $\phi_{exp}$ were calculated as $\phi_{exp} = \sigma_{exp}/\sigma_{model} \cdot \phi_{model}$.

existence of crowding effects at branching points and tips? *(ii)* what proportion of the astrocyte plasma membrane area that could be occupied by glutamate transporters is actually occupied by them? We answered these questions for different sub-cellular compartments, taking into account different insertion depths of glutamate transporters in the plasma membrane.

As a first step, we generated a geometrical approximation for glutamate transporters. These molecules assemble as trimers in the astrocyte membrane, based on information collected from their crystal structure [67]. For this reason, in the model, we approximated each trimer as a triangular prism (S1 Fig). Each trimer in our model was assigned different values of insertion depth, a value that we referred to as $h_{in}$. We initially ignored potential restrictions due to crowding effects at branching points, and we approximated different portions of the astrocyte with simpler 3D morphologies, like spheres (for the soma) and cylinders of different size (for stems, shafts and tips/leaves). The maximum number of transporters that can be placed in a sphere of radius $r_1$, assuming for a moment that these were the only molecules on the plasma membrane, is provided by **Eq (S12)**. We then updated our computations to account for crowding, according to the branching angle and local geometry at each branching point.

Each one of the $K$ primary branches of diameter $\delta_1$ departing from the soma covers a portion of the sphere surface that can be estimated as:

$$A_{cap} = 2\pi r_1 \left( r_1 - \sqrt{r_1^2 - \left(\delta_1/2\right)^2} \right)$$

This means that the effective surface area of the soma where the transporter trimers can be expressed is $A_{soma} - K \cdot A_{cap}$. Based on these considerations, the maximum number of trimers that can populate the soma can be expressed as:

$$n_{soma} = \frac{A_{soma}}{4\pi r_1^2} \cdot n_{sphere} = \frac{4\pi r_1^2 - K \cdot A_{cap}}{4\pi r_1^2} \cdot n_{sphere} \tag{10}$$

where $n_{sphere}(r_1)$ is given in **Eq (S12)**.

The maximum area occupied by all trimers in the soma can be expressed as:

$$S_{soma} = n_{soma} A_{sphere}^{\Delta},$$

where the area of the spherical portion replaced by each trimer $A_{sphere}^{\Delta}(r_1)$ is provided by **Eq (S14)**.

For the astrocyte branches, the total surface area $A_{tree}$, the maximum number of transporter trimers $n_{tree}$ that it can contain, and the total surface area that can be occupied by them $S_{tree}$

were estimated as follows:

$$A_{tree} \quad = \quad \sum_{j=1}^{B} \frac{\pi \delta_j^2}{4} l_j \tag{11}$$

$$n_{tree} \quad = \quad \sum_{j=1}^{B} n_{cyl}\left(\frac{\delta_j}{2}, l_j\right) \tag{12}$$

$$S_{tree} \quad = \quad \sum_{j=1}^{B} n_{cyl}\left(\frac{\delta_j}{2}, l_j\right) A_{cyl}^{\Delta}\left(\frac{\delta_j}{2}\right) \tag{13}$$

where the sums are calculated over all $B$ branch segments $1 \le j \le B$, with lengths $l_j$ and diameters $\delta_j$, computed as specified in the Materials and methods section (the average lengths for the first branching levels are shown in Table 3 as the value of $\lambda$ and the vector $\chi$). Recall that the expression for $A_{cyl}^{\Delta}(r_1)$ was provided by **Eq (S19)**. A combination of our formulas gives us the estimate for the total surface area of the astrocyte:

$$A_{total} = A_{soma} + A_{tree}$$

the maximum number of transporter trimers that it can contain:

$$n_{total} = n_{soma} + n_{tree}$$

and the maximum fraction the astrocyte surface area that can be occupied by them:

$$S_{total} = S_{soma} + S_{tree}$$

We then calculated the maximum transporter trimer surface density as:

$$\sigma = \frac{n_{total}}{A_{total}}$$

The maximum surface density of glutamate transporter monomers was then obtained by multiplying $\sigma$ by three, and the fraction of the astrocyte surface area occupied by the transporters was calculated as:

$$\Phi = \frac{S_{total}}{A_{total}}$$

We computed all these values for a set of model astrocytes generated using MATLAB simulations, with structural features that were in reasonable agreement with those of biocytin-filled astrocytes reconstructed after proExM and 2P-LSM imaging (Fig 4B). If transporters were the only molecules expressed on the astrocyte plasma membrane, crowding effects would pose an upper limit to their surface density in the tips/leaves and, to a lesser extent, in the shafts. These effects would become more pronounced as the transporters protrude towards the cytoplasm, as it happens when they are fully bound [82]. Consequently, the maximum surface density of glutamate transporters would be lowest in the branch tips, and progressively larger in the leaves, shafts, stems and soma, as the size of these sub-cellular compartments increases (Fig 5E). Increasing the insertion depth of transporters at the tips/leaves would reduce their maximal surface density in all compartments (Fig 5E). This would lead to a reduction in the maximum astrocyte surface area occupied by the transporters of $\sim$ 9% in the tips/leaves, and $\sim$ 5% in the shafts, with a much smaller effect in the soma and stems (Fig 5F). By taking

into account the experimental measures of GLAST and GLT-1 surface density, and existing gradients in their expression across different sub-cellular compartments, we estimated their relative surface density ($\sigma_{exp}/\sigma_{model}$; Fig 5G) and effective surface fraction ($\phi_{exp}$; Fig 5H). The results highlight prominent age-dependent effects, whereby glutamate transporters are present at $\sim$ 14% of their theoretical maximal density in astrocyte tips/leaves of 2 week old mice. This value reaches $\sim$ 100% in 6 month old mice, suggesting that at this age, almost all of the surface area of the astrocyte tips/leaves is occupied by glutamate transporters (see Discussion). In other—and larger—sub-cellular compartments, the relative glutamate transporter density is much lower ($\sim$ 3%), and shows a similar age-dependence to the one described for the tips/leaves. Given the large density of these molecules, local changes in the insertion depth of the transporters along the plasma membrane, and therefore conformational changes of these molecules, have small effects on the relative surface density and the effective fraction of the plasma membrane occupied by the transporters. Together, these findings indicate that crowding effects can in principle limit the local expression of membrane proteins like glutamate transporters in astrocytic membrane, although the exact extent to which this happens ultimately depends on the local expression of glutamate transporters and other proteins expressed in the cell membrane of a given sub-cellular compartment.

## Varying the glutamate transporter surface density alters peri- and extra-synaptic glutamate receptor activation

The findings described so far prompted us to ask whether local differences in the expression of glutamate transporters might be of particular physiological relevance for regulating glutamate receptor activation at hippocampal synapses. To address this, we generated a 3D reaction-diffusion Monte Carlo simulation (Fig 6). Here, we distributed seven synapses in a hexagonal array, 500 nm away from each other for consistency with previous anatomical data [5]. We flanked four of them (i.e., 57% [4]) with astrocytic processes covering 43% of their perimeter [4], and released glutamate from the center of the synaptic cleft of the central synapse (Fig 6A–6B). We performed distinct sets of simulations in which we varied the surface density of glutamate transporters in the glial membranes in the range 0–20,000 $\mu m^{-2}$. Here, each transporter molecule could interact with a diffusing glutamate molecule, to bind and/or translocate it across the membrane. The kinetic rates of the binding and translocation reactions were set in accordance to a simplified kinetic scheme for glutamate transporters which does not require an explicit treatment of the movement of co-transported ions (see Materials and methods) [7, 9]. We calculated the probability for each transporter of being in the outward-facing unbound state (To; *magenta*), the outward-facing bound state (ToG; *yellow*) or the inward-facing bound state (TiG; *green*; Fig 6B). The higher the surface density of glutamate transporters, the higher the glutamate uptake capacity of astrocytes. Therefore, the proportion of transporters in each of the three states decreased for higher values of the glutamate transporter surface density (Fig 6C–6D). An increase in astrocytic glutamate transporter expression did not lead to changes in the glutamate concentration profile in the center of the synaptic cleft, above the PSD region (Fig 6E). This is consistent with previous experimental and modeling works, showing that dilution by diffusion dominates the termination of synaptic glutamate receptor activation in the cleft of small central synapses, whereas astrocyte transporters are responsible for limiting spillover activation of glutamate receptors at a distance from the release site [5, 7, 34, 39, 83, 84]. This was also confirmed by our model where the peri/extrasynaptic glutamate concentration profile, which is smaller and arises and decays more slowly than in the cleft [5–7, 34, 39, 83, 84], became progressively smaller as the glutamate transporter concentration in astrocytic membranes increased (Fig 6E–6F) This led to a modest decrease in the open probability of

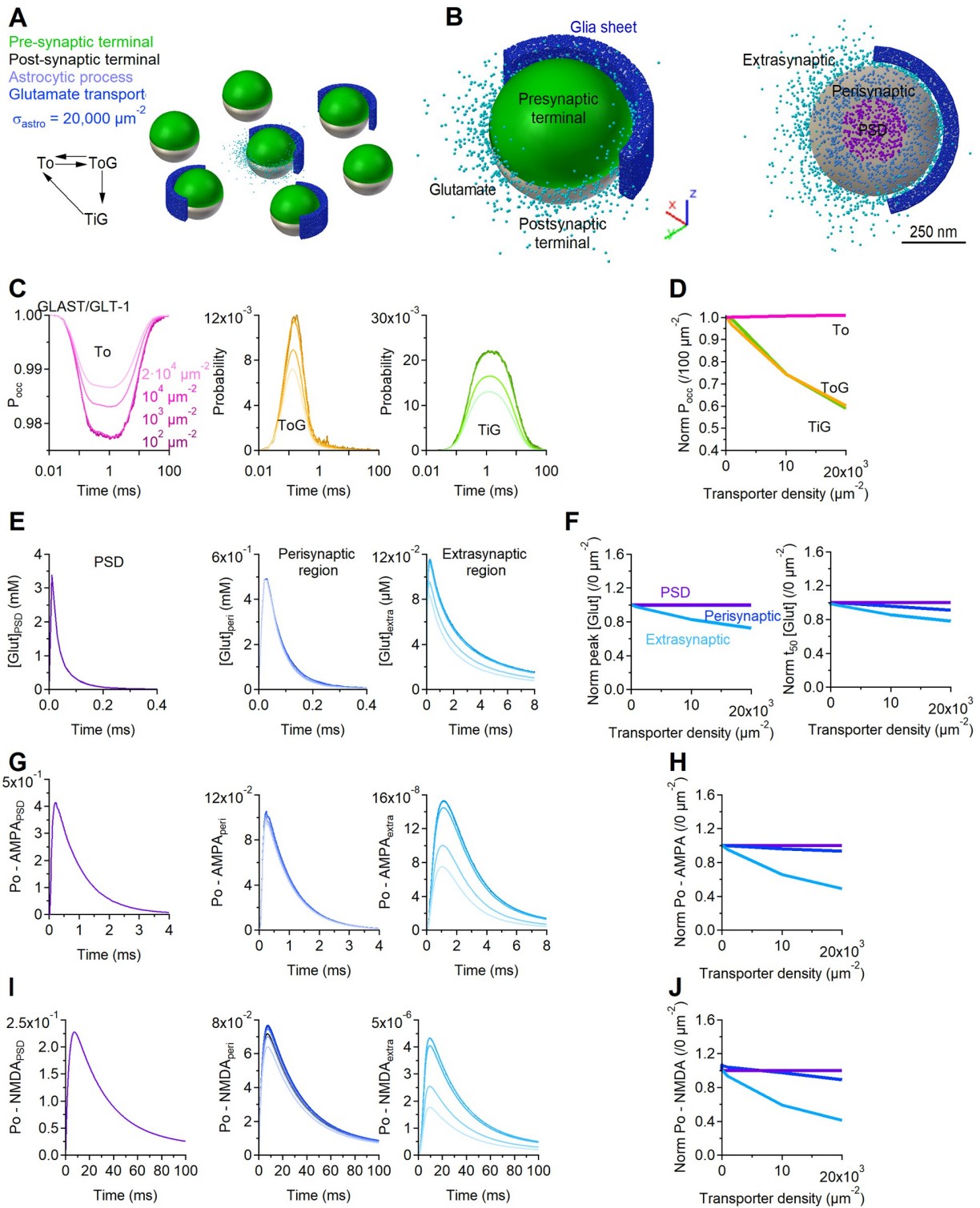

**Fig 6. Varying the glutamate transporter surface density alters extra-synaptic receptor activation. A**. 3D representation of the geometries of seven neighboring synapses used to run the 3D Monte Carlo diffusion model. The kinetic scheme of glutamate transporters, on the left hand side, has the following kinetic: $k_{To} \rightarrow k_{ToG} = 18 \cdot 10^6\ M^{-1}\ s^{-1}$; $k_{ToG} \rightarrow k_{To} = 3,594\ s^{-1}$; $k_{ToG} \rightarrow k_{TiG} = 6 \cdot 10^3\ s^{-1}$; $k_{TiG} \rightarrow k_{To} = 150\ s^{-1}$. **B**. *Left*, Close-up view of an individual synapse, composed of a hemispheric pre- (*green*) and post-synaptic terminal (*white*), with 57% of its perimeter surrounded by an astrocytic process, populated with glutamate transporters (*blue*). Glutamate molecules can be seen diffusing out of the synaptic cleft (*cyan*). *Right*, Close-up view of the synaptic cleft. Glutamate molecules diffusing in the PSD (*purple*) and perisynaptic portion of

the cleft (*cerulean*), and in the extracellular space out of the synaptic cleft (*cyan*). **C.** The three graphs in this panel provide a description of the probability for each transporter of being in one of the three states *To*, *ToG* or *TiG*. The results are obtained using a range of transporter surface density values color-coded in the legend. Lighter colors represent higher values of $\sigma_{astro}$. **D.** Summary describing how the probability of glutamate transporters of being in one of the three states *To*, *ToG* or *TiG* varies with glutamate transporter surface density. The data are normalized based on the results obtained with $\sigma_{astro} = 100\ \mu\text{m}^{-2}$. **E.** Temporal profile of the glutamate concentration in the PSD (*left*), perisynaptic (*center*) and extrasynaptic regions (*right*), for different levels of $\sigma_{astro}$. **F.** Summary of the peak glutamate concentration measured in the PSD (*purple*) and perisynaptic regions (*blue*), and in the extrasynaptic region (*cyan*). The data are normalized based on the results obtained with $\sigma_{astro} = 0\ \mu\text{m}^{-2}$. **G.** Open probability of synaptic (*left*), perisynaptic (*center*), and extrasynaptic AMPA receptors (*right*) measured using different surface density levels of glutamate transporters. **H.** Summary of the peak open probability of AMPA receptors. The data are normalized based on the results obtained with $\sigma_{astro} = 0\ \mu\text{m}^{-2}$. **I,J.** As in G,H, for NMDA receptors. Simulations were run using $\sigma_{astro}$ values of 0, 10, 100, 1,000, 10,000 and 20,000 $\mu\text{m}^{-2}$. Data represent mean values.

perisynaptic receptors, and a sharp decrease in that of extrasynaptic AMPA (Fig 6G–6H) and NMDA receptors (Fig 6I–6J). Together, these results show that local changes in glutamate transporter expression around a synapse can induce dramatic changes in extrasynaptic receptor activation without major consequences on the activation of synaptic receptors. These findings highlight the importance of gaining accurate estimates of the local surface density of glutamate transporters at the synapse as this, through the modulation of AMPA and NMDA receptor activation, can ultimately shape the temporal dynamics of fast synaptic transmission and the propensity to undergo plastic changes in synaptic strength.

In a last set of simulations, we varied the location of the glutamate release site from the center to the periphery of the PSD, either towards the portion of the synapse facing the astrocyte process or away from it (Fig 7A). These represented the most extreme scenarios, as they covered the whole range of distances that a release site can be: from the closest to the furthest away from the neighboring glial sheet. Clearly, release sites located at intermediate distances would experience more nuanced flavors of the data that we describe here. Changing the surface density of glutamate transporters altered the probability of occupancy of glutamate-bound transporters (inward-facing and outward-facing; Fig 7B), and altered the open probability of AMPA and NMDA receptors (Fig 7C). By contrast, since glial glutamate transporters outnumber the glutamate molecules released in the cleft, the proportion of outward-facing unbound glutamate transporters remained relatively stable. These effects were most pronounced when glutamate release occurred from a release site close to neighboring glial sheets, suggesting that the location of the release site with respect to the glial sheet is crucial to determine the extent to which the glial sheet can limit post-synaptic glutamate receptor activation. The further is the release site from the glial sheet, the more likely it is for glutamate to activate peri/extrasynaptic receptors.

We explored these results more closely by constraining the surface density of glial glutamate transporters to 20,000 $\mu m^{-2}$, similar to their concentration in tips/leaves. We changed the location of the release site from the center of the PSD to the closest and farthest points to the neighboring glial sheet (Fig 8). As noted in (Fig 6), the closer the release site to the glial sheet, the more likely it is for glutamate to be bound by a glial transporter (Fig 8A–8B) and not by extrasynaptic AMPA and NMDA receptors (Fig 8C–8H). Perisynaptic receptors are less susceptible to these effects, but their activation is slightly reduced when the release site is not located at the center of the synaptic cleft. This is because when the release site is closer to the glial sheet, a larger proportion of glutamate is bound by glial transporters. When the release site is farther from the glial sheet, a large portion of glutamate diffuses out in the extracellular space, out of the synaptic cleft. These effects are similar for AMPA and NMDA receptors, and this is not surprisingly given their similar glutamate binding rates [59, 60]. In summary, glutamate spillover from the synaptic cleft is shaped not only by the extent of glial coverage of a given synapse, but also on the identity of the sub-cellular compartment forming that glial coverage, and the relative location of the release site with respect to the glial sheet.

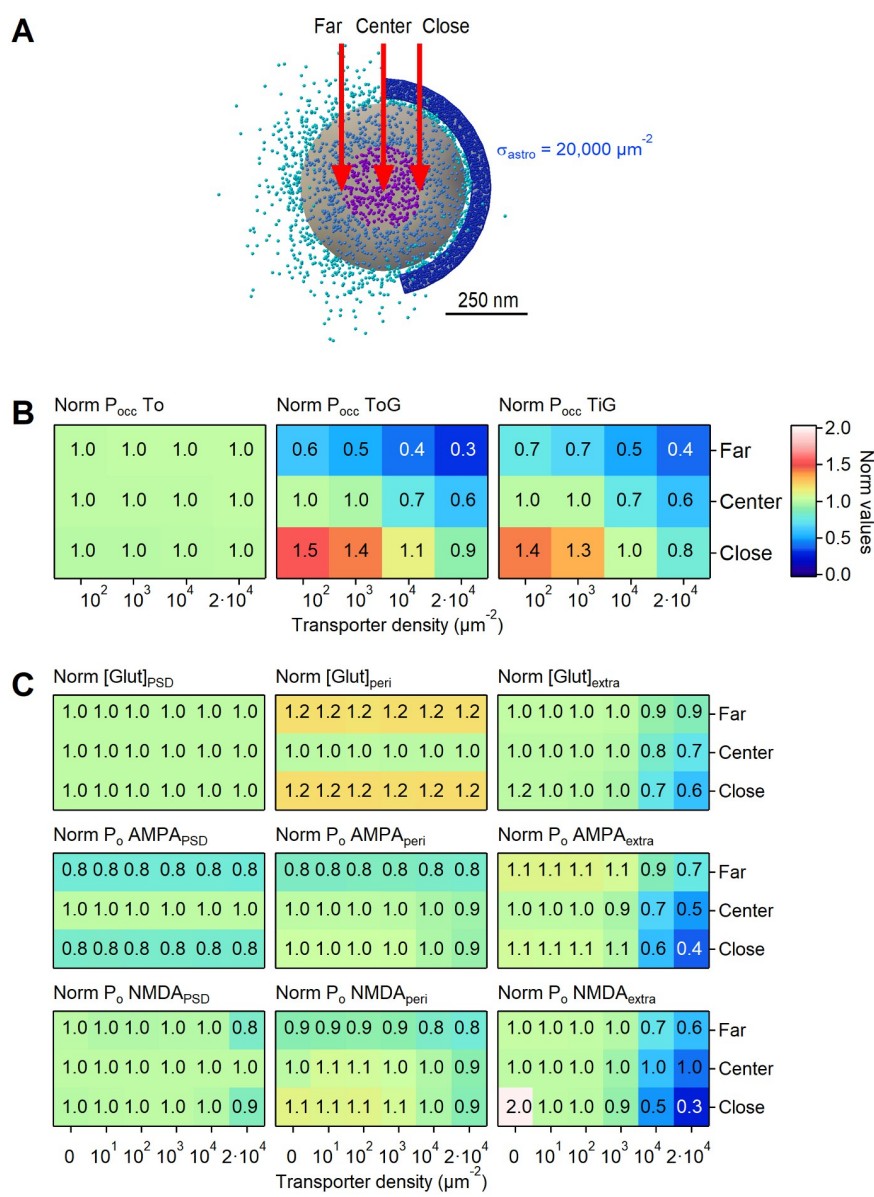

**Fig 7. Varying the glutamate transporter surface density has more profound effects on receptor activation if the release site is located closer to the glial sheet. A**. Close-up view of the synaptic cleft. Glutamate molecules diffusing in the PSD (*purple*) and perisynaptic portion of the cleft (*cerulean*), and in the extracellular space out of the synaptic cleft (*cyan*). The three red arrows point to the location of the glutamate release site in different three sets of simulation. Its location was either *(i)* above the center of the PSD (*center*), or at its edges, *(ii)* close or *(iii)* far from the glial sheet. In this figure, the surface density of glutamate transporters was set to $\sigma_{astro}$ = 20,000 $\mu m^{-2}$. **B**. Summary graph, showing how the probability of occupancy of the To (*left*), ToG (*center*) and TiG states (*right*) varies depending on the location of the glutamate release site. The data are normalized by the values obtained with $\sigma_{astro}$ = 100 $\mu m^{-2}$ (trough for To and peak for ToG and TiG), and a release site located above the center of the PSD. **C**. Summary graph, showing how the peak glutamate concentration (*top*) and peak open probability of AMPA (*middle*) and NMDA receptors (*bottom*) varies depending on the location of the glutamate release site. The estimates were obtained for the PSD (*left*), perisynaptic (*center*) and extrasynaptic regions (*right*). The data are normalized by the values obtained with $\sigma_{astro}$ = 0 $\mu m^{-2}$, and a release site located above the center of the PSD. For clarity, error bars are not included in this figure.

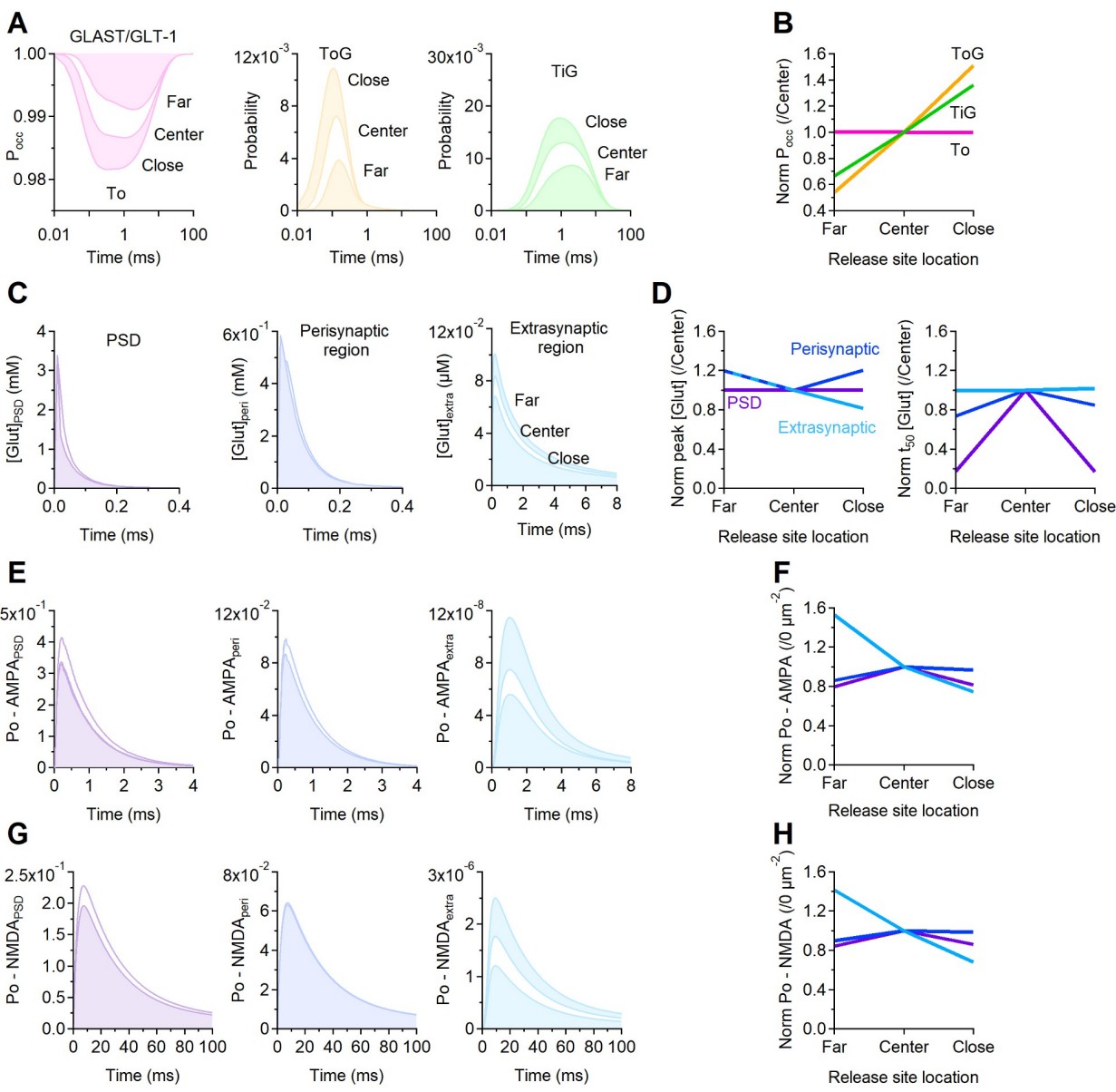

**Fig 8. Changing the location of the release site for $\sigma_{astro}$ = 20,000 $\mu m^{-2}$. A.** The three graphs in this panel provide a description of the probability for each transporter of being in one of the three states $To$, $ToG$ or $TiG$, with $\sigma_{astro}$ = 20,000 $\mu m^{-2}$, when the release site is located above the center of the PSD, close or far from the glial sheet (Fig 6A). **B**. Summary describing how the probability of glutamate transporters of being in one of the three states $To$, $ToG$ or $TiG$ varies with release site location. The data are normalized based on the results obtained when the release site is located above the center of the PSD. **C**. Temporal profile of the glutamate concentration in the PSD (*left*), perisynaptic (*center*) and extrasynaptic regions (*right*), for different release site locations. **D**. Summary of the peak and half decay time ($t_{50}$) of the glutamate concentration measured in the PSD (*purple*) and perisynaptic regions (*blue*), and in the extrasynaptic region (*cyan*). The data are normalized based on the results obtained with the release site located above the center of the PSD region. **E**. Open probability of synaptic (*left*), perisynaptic (*center*), and extrasynaptic AMPA receptors (*right*) measured for different release site locations. **F**. Summary of the peak open probability of AMPA receptors. The data are normalized based on the results obtained with $\sigma_{astro}$ = 0 $\mu m^{-2}$. **G,H**. As in E,F, for NMDA receptors. Lines are used in panels B, D, F, H to facilitate the interpretation of the results, collected using three different release site locations. Data represent mean values.

## Discussion

### Comparison of present and past findings in rats

Currently available estimates of glutamate transporter density in astrocytic membranes, obtained from 7–8 week old rats [11], have been widely used in modeling studies of glutamate diffusion, uptake and cross-talk among synapses for different rodent species and ages [6, 7, 32, 33, 36, 52, 57, 64, 83, 85, 86]. These estimates showed that the total concentration of glutamate transporters in *stratum radiatum* of the rat hippocampus is $\sim 30\ \mu$M [11], slightly lower than what we obtained in our own experiments ($\sim 58\ \mu$M). There are multiple methodological differences between our experiments and the ones of [11] that could account for this minor discrepancy. *First*, we use different protein extraction methods. Lehre and Danbolt homogenized their whole hippocampal tissue with 1% SDS, according to methods described by [87–89]. We used a membrane protein extraction kit to separate membrane-bound from cytoplasmic proteins. If anything, this method could lead to a slight underestimate of our measures, if some membrane-bound proteins were lost in the cytoplasmic fraction. However, we think this is an unlikely scenario as our experiments did not reveal detectable signals for GLAST in the cytoplasmic fraction of our samples, and only $\sim 0.2\%$ for GLT-1. *Second*, Lehre and Danbolt raised their own antibodies against GLAST and GLT1, whereas we used commercially-available ones. There could be unknown differences in their sensitivity and specificity. *Third*, there might be slight differences in reactivity of our GLAST antibody to glutamate transporters from different species. The GLAST antibody used in this work was raised against a human antigen; while the exact antigen sequence is proprietary, this region of the protein is 93% identical in mice and 95% identical in rats. However, we do not anticipate species recognition to be an issue for the GLT-1 antibody, because the antigen sequence is 100% identical in humans, rats and mice. *Fourth*, there might be subtle differences in the protein concentrations of GLAST and GLT-1 across rat strains, since previous work used Wistar rats and our work used Sprague Dawley rats.

Despite these potential differences in protein concentration, our surface density measures for GLAST and GLT-1 are in agreement with the values obtained by [11]. This is due to the fact that the conversion from protein concentration to surface density requires the use of a numerical estimate for the astrocyte surface density. This value was 1.4 $\mu m^2/\mu m^3$ for [11] and 3.15 $\mu m^2/\mu m^3$ for our work. The value of 1.4 $\mu m^2/\mu m^3$ was selected by [11] for being in reasonable agreement with estimates of the astrocyte surface density in layers I-IV of the visual cortex of male Long Evans rats aged P23–25 (i.e., 1.32–1.64 $\mu m^2/\mu m^3$) [12]. These values, however, were obtained from the analysis of collections of 2D EM images, which does not provide a direct readout of astrocyte membranes in 3D. Our estimates, by contrast, are based on the analysis of 3D EM reconstructions of tissue samples from hippocampal *stratum radiatum* of P17 mice. Given that the proportion of neuronal versus non-neuronal cells, and the shape and size of non-neuronal cells is very similar in the neocortex of mice and rats [74], we chose to use 3.15 $\mu m^2/\mu m^3$ as the astrocytic surface density for all samples analyzed in this work. We acknowledge that these calculations rely on a number of approximations and that there are limitations to the experimental approaches used; however, we believe they provide reasonable ballpark figures for the distribution of glutamate transporters in the hippocampal neuropil.

Despite the technical advantages provided by proExM, which allows to literally make small structures bigger with isotropic expansion, this approach also lead to a dilution of target molecules or fluorophores [51]. This is simply due to the fact that now the same number of molecules occupy a larger volume. A consequence and potential drawback of this effect is that very small structures can fall below the detection limit of 2P-LSM. There is evidence that the cellular architecture of astrocytes is complex and finer than what can be distinguished using images

approaches with a spatial resolution of a few hundred of nanometers [64–66, 76, 90–92]. Therefore, the 3D axial STEM tomography analysis provides an ideal complement to the information that can be collected from proExM and 2P-LSM imaging of biocytin-filled astrocytes.

## Age-dependent changes in glutamate transporter expression in mice

As mice have become widely used in the scientific community, it is important to determine whether the estimates collected from rats provide a good approximation of the glutamate transporter concentration and surface density in astrocytes of the mouse hippocampus. Our results show that the concentration of glutamate transporter is similar between 7–8 week old mice and rats ($\sim$ 76 ± 11 versus $\sim$ 61 ± 11 $\mu$M, n = 3; p = 0.091). These values, however, change with mouse age, being 17 $\mu$M in 2 week old mice, increasing to 98 $\mu$M in 6 month old mice, and then declining to 66–80 $\mu$M in 15–21 month old mice (Table 4). Since the extracellular space in the mouse hippocampus is $\sim$ 15% [32], the effective concentration of glutamate transporters in portions of the extracellular space delimited by astrocytic membranes is likely in the order of 110 − 650 $\mu$M within the mouse age range studied here. This is consistent with values previously obtained from rats (cf. 140–330 $\mu$M in rats [11]). Assuming that the total plasma membrane density is 14 $\mu$m$^2$/$\mu$m$^3$ [5, 11], then the fraction of the extracellular space enclosed between an astrocyte membrane and any other membrane is $\sim$ 35% in mice (i.e., 2 · 3.15/14 · (14 − 3.15)/14 $\mu$m$^2$/$\mu$m$^3$). This means that in the mouse hippocampus, the effective concentration of glutamate transporters in portions of the extracellular space flanked by astrocytic and non-astrocytic membranes is 0.3 − 1.9 mM, depending on the mouse age.

These values, which are similar to those previously obtained from rats, indicate that the glutamate uptake capacity in the hippocampus can vary substantially with age: increasing progressively until adulthood and then starting to decline after 6 months of age. We believe our data extend previous works through the analysis of a much broader age window. For example, developmental changes in postnatal expression of GLAST and GLT-1 in the mouse hippocampus were previously analyzed between 3–5 days and 2 months of age [44]. According to this work, in mice, the relative expression of GLAST increases progressively during the first three post-natal weeks, when it plateaus to levels found in adult tissue (i.e., 2 month old) [44]. By contrast, the expression of GLT-1 continues to increase progressively between 3–5 days after birth and 2 months of age in mice [44]. Our results differ from those of [44], which suggested that the expression of GLAST reaches adult protein levels at P20–25. This discrepancy might be simply due to the fact that [44] did not analyze mice older than 2 months, possibly overlooking a further increase in GLAST expression that occurs at a later time point. We find that the developmental trend of expression of GLAST and GLT-1 within the wide range of ages tested here (from 2 weeks to 21 months) is qualitatively similar. In both cases, the expression of these transporters continues to increase until 6 months of age. This is consistent with the work of [93] in rats, showing a progressive increase in the postnatal expression of GLAST and GLT-1 until adulthood in the cortex, cerebellum and striatum, albeit with different magnitude and time course across brain regions. However, the exact age of the adult rats used by [93] is not included in their work, making a close comparison particularly challenging. A key difference that we find between GLAST and GLT-1 lies in the fact that the surface density of GLT-1 is 0.85 times that of GLAST in 2 week old mice, and increases to 2 times that of GLAST in 3 week old mice. In 6 month old mice, the surface density of GLT-1 is 3 times that of GLAST. Therefore, GLT-1 is already the dominant glutamate transporter variant in the hippocampus of 3 week old mice, well before adulthood. The different levels of expression of GLAST and GLT-1 suggest the existence of potential differences in molecular and cellular mechanisms that regulate GLAST and GLT-1 expression in mouse astrocytes throughout development. These

regulatory mechanisms include complex signaling cascades and suggest that different glutamate transporter variants might have different susceptibility to their activation [94].

In this general context, our data indicate that the expression of the astrocyte glutamate transporters GLAST and GLT-1 increases progressively from adolescence to young adulthood and middle age, before declining with aging. This increased expression could function to preserve the spatial specificity of synaptic transmission, limiting the ability of glutamate to activate receptors at a distance from its release site. It could also provide a mechanism to reduce further the steady-state concentration of glutamate in the extracellular space, with potentially notable effects on fast synaptic transmission, short- and long-term plasticity, as well as on the dynamics of biosynthetic mechanisms to regulate glutamate biosynthesis and degradation in astrocytes.

On the other hand, a disruption of these regulatory mechanisms of glutamate homeostasis could act as a main contributing factor to cognitive decline and neurological disease in the aging brain. A natural decline in the expression of astrocytic glutamate transporters has been suggested to occur in rats ($3 - 6$ month versus $24 - 27$ month old) [95], 3 month versus 24 month old [96]) and also in humans ($8 - 40$ versus and $40 - 63$ years old [97]). In humans, a decline in glutamate transporter expression has also been detected in neurodegenerative disease like attention deficit hyperactivity disorder [98, 99], autism [98, 100], depression [101–103] and Alzheimer's disease [104–107], prompting the idea that it may represent an important risk factor for these disease [108].

## Uneven distribution of glutamate transporters along the plasma membrane of astrocytes

Glutamate transporters are unevenly distributed along the astrocytic plasma membrane [13, 16, 42]. Accordingly, immunostaining experiments in hippocampal sections and organotypic slice cultures show that GLT-1 forms small clusters in lamellae and filopodia-like processes, the size and distribution of which can be modulated by neuronal activity [42]. This confined expression, which is a common property among transporters with trimeric assembly like ASCT1, is not attributed to the presence of consensus sequences in the cytoplasmic tails of the transporters, which act as binding sites of cytosolic scaffolding proteins. Instead, it has been suggested to be due to the presence of an extracellular loop at the center of the $4^{th}$ transmembrane helices, not far from the glutamate binding sites, which does not require transporter activity [48]. In the astrocyte tips at the end of astrocyte branches, and in fine leaves protruding laterally from each branch, the surface density of glutamate transporters is high, due to the presence of a transmembrane domain acting as a sorting signal to the astrocyte processes tips [48]. The increased density of glutamate transporters in astrocytes tips is largely due to a 2.3-fold increase in GLT-1 expression in the tips/leaves compared to the soma [48, 75]. By contrast, GLAST is 0.6-fold less abundant in the tips/leaves compared to the soma. These findings indicate that the molecular mechanisms responsible for sorting glutamate transporters to different sub-cellular compartments might differ among specific variants [48, 75]. Our data show that these tips, defined as the hemispheric cap of terminal processes, together with leaves, represent a relatively small portion of the surface area of an astrocyte ($8.7 \pm 2.7\%$). A typical astrocyte occupies a 588,000 $\mu m^3$ domain of the hippocampus (Fig 4E). Within this volume, there are $588,000 - 1,764,000$ synapses, based on anatomical estimates suggesting that there are $1 - 3 \ \mu m^{-3}$ synapses in the hippocampal neuropil [109]. Given that only 57% of these synapses have an astrocytic process apposed to them [4], the number of synapses contacted by an astrocyte is $335,000 - 1,005,000$. This number is higher than the average number of tips/leaves of an astrocyte (2,456), suggesting that only $0.2 - 0.7\%$ of all synapses in the neuropil volume

occupied by an astrocyte are contacted by a tip/leaf. For GLAST, the tips/leaves:shaft:stem: soma expression ratio with respect to the soma is 0.6:0.5:0.8:1, whereas this is 2.3:1.4:0.7:1 for GLT-1. In other words, the tips/leaves:shaft:stem:soma expression ratio with respect to the soma of all glutamate transporters is 1.4:0.9:0.7:1. This means that $\sim$ 99% synapses contacted by an astrocyte have only $50 - 71\%$ of the glutamate transporters found at synapses contacted by astrocyte tips/leaves. At first sight, these estimates might seem at odds with those collected in anatomical studies showing that in layer II of the mouse somatosensory and anterior cingulate cortex, $\sim$ 60% of the astrocytic processes in contact with excitatory synapses come from leaflets, $\sim$ 37% from branchlets and $\sim$ 3% from branches [110]. This classification was based on the diameter of these processes [111]. Here, the term leaflets refers to processes with dimensions on the scale of hundreds of nanometers [111]. Our classification is slightly different, as it is based on the identity of a process. Although the diameter of tips/leaves is $\sim$ 30 nm, stems and branches are defined as such regardless of their dimensions, and these can vary significantly as one moves from the base to the ending of a process. For this reason, we do not think that our results and those of [110] might be necessarily in conflict. Unfortunately, some level of ambiguity in the terminology pervades the field, and this might be due, in part, to historical limitations of 2D-EM works, where determining the identity of the astrocytic sub-cellular compartment in contact with a synapse was a particularly challenging task. While trying to generate a unifying terminology is desirable but beyond the scope of this work, we emphasize the importance of providing clear definitions when it comes to the nomenclature of different sub-cellular compartments. Since changes in the local density of glutamate transporters alter the extent of extrasynaptic AMPA/NMDA receptor activation, these findings indicate that there is a significant proportion of synapses where the activation of these receptors is favored. Our findings also indicate that crowding effects my introduce an upper limit to the expression of glutamate transporters in spatially confined astrocytic processes. This, in turn, could contribute to the limited surface mobility of these molecules in astrocyte tips [2].

At first sight, the findings that glutamate transporter occupy essentially all the surface area of tips/leaves may in astrocytes of 6 month old mice might seem at odds with transcriptomics data detecting mRNAs for molecules other than glutamate transporters that can be readily transcribed in peripheral or perisynaptic astrocytic processes (e.g., potassium channels, connexins) [112, 113]. Note, however, that in these works, these astrocytic processes are identified for their proximity to synapses, but can be formed by any of the sub-cellular compartments described in our own work (i.e., soma, stems, shafts, tips/leaves). For this reason, we do not think that there is any inconsistency between our and previous transcriptomic data. Last, we would like to point out that our data provide ballpark figures. As such, the finding that 100% of the surface area of astrocyte tips/leaves in 6 month old mice is occupied by transporters indicates that the vast majority of the surface area of these processes is occupied by transporters. The possibility that other membrane proteins might be expressed is not ruled out, but just seems rare according to our estimates.

## The 3/2 rule applies to astrocytes

According to Rall's equivalent cylinder model, the complexity of branching geometries of cells can be reduced to that of a simple cylinder [61]. This condition is satisfied, in neurons, when the relationship between the diameter of the parent and daughter branches at a particular branching point follows a constraint commonly referred to as the 3/2 rule [61–63]. Under these conditions, the input conductance of dendritic branches depends on the 3/2 power of the branch diameter [61]. Our anatomical measures indicate that the diameter distribution of daughter and parent astrocytic branches at a branching point also appear to be described by

the 3/2 rule (Table 3). The ability to map the entire branching geometry of an astrocyte as an equivalent cylinder according to the 3/2 rule means that the input conductance of each branch cylinder also depends on the 3/2 power of its diameter and that the fundamental rules of passive signal propagation and integration through astrocyte branches share the same biophysical properties of neurons, validating the use of compartmental models for these cells.

## Conclusion

Together, our findings provide a quantitative framework to study glutamate uptake and signal propagation in astrocytes, with long-range effects on the control of synaptic specificity. We provide estimates of glutamate transporter surface density in different sub-cellular compartments in astrocytes of the mouse hippocampus, at different ages. Strikingly, the identity of the sub-cellular domain of the astrocyte membrane that is in contact with a synapse emerges as a major determinant of glutamate spillover and extrasynaptic receptor activation, due to the uneven distribution of glutamate transporters. Recent findings identify local hot spots for astrocytic membrane depolarization induced by extracellular potassium accumulation during action potential generation, which might alter the efficacy of glutamate uptake through their action on glutamate transporters [114]. These findings may point out to the existence of local clusters of potassium channels on the astrocyte membrane, although their distribution along different sub-cellular compartments remains to be determined. The emerging scenario, based on this and our own work, is that of a complex molecular landscape of the astrocyte plasma membrane, a far cry from the assumptions of uniform molecular distributions that are presently used.

## Supporting information

**S1 Appendix. Geometrical computations.** This file contains the mathematical details of for estimating the maximal density of glutamate transporters in a circle, sphere and cylinder.
(PDF)

**S1 Fig. Geometrical approximations used for modeling transporters. A**. Extracellular view of the ribbon representation of a $Glt_{Ph}$ trimer, with protomers shown in orange, magenta and cyan. The green triangle with side length $w = 8$ nm is used to approximate the cross section of the trimer. **B**. View of the trimer parallel to the membrane. The green rectangle with $h = 6.5$ nm height is used to approximate the side view of the trimer. **C**. Spherical triangle (formed by three intersecting lunes) with spherical angles $\alpha$, $\beta$ and $\gamma$, corresponding to angles at the center of the sphere called $a$, $b$ and $c$, respectively. **D**. Schematic representation of a trimer (green triangular prism) protruding through the plasma membrane of a spherical compartment. **E**. Schematic representation of a transporter protruding inside a cylindrical compartment; the bottom of the trimer and its projection onto the $x$, $y$ Cartesian coordinate plane are shown as green triangles. **F**. Projection of the trimer onto the coordinate plane, used as the domain for computing the double integrals for $A_{sphere}^{\Delta}$ and $A_{cyl}^{\Delta}$.
(TIF)

**S2 Fig. Spatial relationship between adjacent glutamate transporter trimers in a circle. A**. Each transporter trimer had a portion of length ($h_{in}$) that could protrude inside the circle from the cytoplasmic layer of the plasma membrane. The side length of the trimers was $\omega = 8$ nm. This spatial arrangement provided a framework to estimate the maximum number of transporter trimers and monomers that can reside in the perimeter of a circle ($n_{circle}$) of radius $r_1$. **B**. Schematic representation of circles with transporter trimers. The glutamate transporter trimers were represented as rectangles with 8 nm side length and 6.5 nm height. The example at

the top refers to trimers that do not protrude into the circle. In the example in the middle, the trimers were located half way through the plasma membrane and protruded in the circle by 1.75 nm. In the case described at the bottom, the trimers protruded all the way through the membrane, for 3.5 nm. **C**. Estimates of the maximum number of transporter molecules ($3n_{circle}$) that can be positioned along the circumference of circles of different radius $r_1$. We examined the three different cases described in (A), where $h_{in} = 0$ nm (*black*), $h_{in} = 1.75$ nm (*dark green*) and $h_{in} = 3.5$ nm (*light green*). **D**. Estimates of the 2D glutamate transporter monomer density for each value of $h_{in}$. **E**. Proportion of the circle perimeter occupied by transporter monomers for increasing values of the circle radius. An asymptotic value of 1 was reached at varying rates depending on the value of $h_{in}$.
(TIF)

**S3 Fig. Spatial relationship between adjacent glutamate transporter trimers in a sphere. A**. *Top*, Spatial distribution of transporter trimers in a sphere. *Bottom*, A spherical triangle is a figure formed on the surface of a sphere by three great circular arcs intersecting pairwise in three vertices. The spherical triangle is the spherical analog of the planar triangle. The figures shows a spherical triangle with angles $a$, $b$, $c$, equivalent to the angles $\alpha$, $\beta$, $\gamma$. **B**. Schematic representation of spheres used as 3D representations of the astrocyte soma. Glutamate transporter trimers are represented as triangular prisms with a base of 8 nm side length and 6.5 nm height. The example at the top refers to trimers that do not protrude into the sphere. In the example in the middle, the trimers are located half way through the plasma membrane and protrude in the sphere by 1.75 nm. In the case described at the bottom, the trimers protrude all the way through the membrane, for 3.5 nm. **C**. Estimates of the maximum number of transporter trimers $n_s phere$ that can be positioned along the surface of spheres of different radius $r_1$. We examined the three different cases described in (B), where $h_{in} = 0$ nm (*black*), $h_{in} = 1.75$ nm (*dark green*) and $h_{in} = 3.5$ nm (*light green*). **D**. Estimates of the 3D transporter trimer surface density for each value of $h_{in}$. **E**. Fraction of the surface area of the sphere that can be occupied by transporter trimers.
(TIF)

**S4 Fig. Spatial relationship between adjacent glutamate transporter trimers in a cylinder. A**. Schematic representation of trimer arrangement around the lateral surface of a cylinder. For clarity, we only show three adjacent trimers, but in reality they keep covering the entire lateral surface of the cylinder using the same pattern. **B**. Schematic representation of cylinders used as 3D representations of astrocytic branches. Glutamate transporter trimers are represented as triangular prisms with a base of 8 nm side length and 6.5 nm height. The example at the top refers to trimers that do not protrude in the lumen of the cylinder. In the example in the middle, the trimers are located half way through the plasma membrane and protrude inside the cylinder by 1.75 nm. In the case described at the bottom, the trimers protrude all the way through the membrane, for 3.5 nm. **C**. Estimates of the maximum trimer number $n_{cyl}$ that can be positioned along the lateral surface of cylinders of different radius $r_{cyl}$ (the height of the cylinder in this case is set to $L = 1$ $\mu$m. We examined the three different cases described in (B), where $h_{in} = 0$ (*black*), $h_{in} = 1.75$ (*dark green*) and $h_{in} = 3.5$ nm (*light green*). **D**. Estimates of trimer surface density for each value of $h_{in}$. **E**. Estimates of the fraction of the lateral surface of the cylinder that can be occupied by trimers for each value of $h_{in}$. The area shaded in magenta represents the range of experimental measures for tip diameters.
(TIF)

**S5 Fig. Geometric constrains on trimer distribution close to branch points. A**. The illustration shows in cross section the astrocyte geometry at a branch point and at a branch tip. Each

transporter trimer is represented as a green rectangle of height $h$. In this specific illustration, the trimer is protruding out of the plasma membrane by $h_{out}$, but our calculations were performed for $h_{in} = 0$–3.5 nm. The branching points and branch tip are regions that display a crowdedness effect, which reduces the local density of transporter trimers (as further explained in the text). **B**. Close-up view of the tip of an astrocyte process ending either as a cylinder (*left*) or as a hemisphere (*right*). **C**. Estimates of the maximum trimer number that can be placed in a cylinder with height equals to the radius (*solid line*) or in a hemisphere (*dashed line*), for $h_{in} = 0$ nm (*top*), $h_{in} = 1.75$ nm (*middle*) and $h_{in} = 3.5$ nm (*bottom*). **D**. As in C, for trimer density. **E**. As in C, for the portion of the plasma membrane occupied by transporter trimers.
(TIF)

**S6 Fig. Walking cubes approximation for measuring astrocyte process volume fraction in the gliapil.** Simulated surrounding volume for an example cell with 3 primary branches, using a mesh of cubes with side length of 2 $\mu$m (shown in red). The blue wedge represents a portion of the astrocyte soma. The astrocyte processes departing from the soma are color coded based on their branch level.
(TIF)

## Acknowledgments

We would like to thank Pavel I. Ortinsky for comments on the manuscript, and Richard D. Leapman and Alioscka A. Sousa for help with 3D axial STEM tomography data collection.

## Author Contributions

**Conceptualization:** Anca R. Rădulescu, Annalisa Scimemi.

**Data curation:** Anca R. Rădulescu, Gabrielle C. Todd, Annalisa Scimemi.

**Formal analysis:** Anca R. Rădulescu, Gabrielle C. Todd, Annalisa Scimemi.

**Funding acquisition:** Anca R. Rădulescu, Annalisa Scimemi.

**Investigation:** Anca R. Rădulescu, Cassandra L. Williams, Benjamin A. Bennink, Alex A. Lemus, Haley E. Chesbro, Justin R. Bourgeois, Ashley M. Kopec, Damian G. Zuloaga, Annalisa Scimemi.

**Methodology:** Anca R. Rădulescu, Annalisa Scimemi.

**Project administration:** Annalisa Scimemi.

**Resources:** Annalisa Scimemi.

**Software:** Anca R. Rădulescu, Annalisa Scimemi.

**Supervision:** Annalisa Scimemi.

**Validation:** Annalisa Scimemi.

**Visualization:** Annalisa Scimemi.

**Writing – original draft:** Anca R. Rădulescu, Annalisa Scimemi.

**Writing – review & editing:** Anca R. Rădulescu, Gabrielle C. Todd, Annalisa Scimemi.

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
