## [Decision Letter · Decision Letter 0]

7 Jul 2021

Dear Dr Scimemi,

Thank you very much for submitting your manuscript "Estimating the glutamate transporter surface density in mouse hippocampal astrocytes" for consideration at PLOS Computational Biology.

As with all papers reviewed by the journal, your manuscript was reviewed by members of the editorial board and by several independent reviewers. In light of the reviews (below this email), we would like to invite the resubmission of a significantly-revised version that takes into account the reviewers' comments.

We cannot make any decision about publication until we have seen the revised manuscript and your response to the reviewers' comments. Your revised manuscript is also likely to be sent to reviewers for further evaluation.

Sincerely,

Hugues Berry

Associate Editor

PLOS Computational Biology

Lyle Graham

Deputy Editor

PLOS Computational Biology

Reviewer's Responses to Questions

**Comments to the Authors:**

Reviewer #1: In this manuscript, Radulescu et al. estimate the surface density of glutamate transporters at the plasma membrane of astrocytes, highlighting differences depending on the sub-cellular compartment, species and mouse age. To do so, they performed dot blotting on protein extracts from mouse hippocampus as well as calculations. Stochastic reaction-diffusion simulations of glutamate uptake in a cube world containing 7 adjacent synapses predict the effect of the local amount of glutamate transporters on the activation of post-synaptic glutamate receptors. Further, 2-photon excitation laser scanning fluorescence microscopy on brain slices treated with protein retention expansion microscopy protocol is used to characterize astrocyte morphology, allowing the generation of 3D reconstructions of astrocytes. Based on the resulting morphological analysis, the authors propose a custom-made code that allows the generation of a bank of 3D models of astrocytes that can be used for in silico simulations and for morphological analysis. Finally, calculations were performed to investigate the surface area that can be occupied by glutamate transporters onto the astrocytic membrane depending on local morphology, notably cylinder versus sphere, thus proposing the maximum amount of transporters on each cellular sub-compartment. Those calculations suggest that the nature of the astrocytic compartment contacting synapses affects the local density of glutamate transporters. Since glutamate uptake affects synaptic activity, glutamate spillover, and is altered in brain disorders such as epilepsy, better characterizing the localization of its effectors, glutamate transporters, is crucial.

While the question investigated by this study is of tremendous importance in the field, I have some concerns regarding the relevance of the methods proposed to address it and regarding the novelty of the results obtained. I would also suggest major reorganization of the manuscript.

MAJOR POINTS

1. Discussion of the main insights

My first major concern is that, while the manuscript presents an impressive amount of work, the relevance and novelty of the presented results compared to previous works are a bit unclear and not sufficiently discussed. Key recent articles in the field are notably not cited (see details below).

1.a. Distribution of glutamate transporters

Recent electron and super-resolution studies have quantified the density of glutamate transporters with high spatial resolution (see e.g the recent review from Groc & Choquet, Science, 2020 for an overview of the techniques, Ciappelloni et al, Cell Calcium 2017 for single nanoparticle tracking in live astrocytes and quantification of transporter surface diffusion and Heller et al, Methods, 2020 for dSTORM nanoscale localization of individual astrocytic glutamate transporters). Further, Schreiner et al, Journal of Comparative Neurology, 2013, quantified the expression profile of GLAST and GLT-1 in the mouse hippocampus at different postnatal ages: P3–P5; P10–P15; P20–P25 and P60–P70, using immunohistochemistry, western blotting, confocal and STED microscopy. They have reported, similarly to this manuscript, that both GLAST and GLT-1 immunoreactivity increased during postnatal development. Schreiner et al. further used confocal and super-resolution microscopy to characterize the sub-cellular localization of glutamate transporters. This highlights that the distribution of glutamate transporters in mouse hippocampus during development has already been addressed by previous studies. Advances in the field provided by the submitted manuscript compared to the above-mentioned studies should be clarified.

Assumptions made for the calculations presented in this manuscript need to be discussed. For example, spatial constraints dictating trimer distribution onto the surface of each astrocytic sub-compartment should strongly vary with branching angle. Also, there is increasing evidence that EAATs are expressed by other cells than astrocytes (Suarez-Pozos et al, 2020, Neurochem Res) and GLAST can be located on the membrane of the endoplasmic reticulum. This should be discussed.

The study compares the density of glutamate transporters in mouse VS rat by comparing their experimental results measured in mice with a study on rats that is more than 20 years old. The results could differ because of a difference in the protocol/laboratory doing the experiment. To assess the difference between mouse and rat glutamate transporters expression, experiments within the same lab on protein extracts from both rat and mouse hippocampus is required.

1.b. Effect of the density of glutamate transporters on glutamate concentration in the cleft

The model predicts that the density of transporters does not impact the glutamate concentration profile in the synaptic cleft (Fig 3C). This result is quite unexpected and needs to be discussed and compared with results from other models of tripartite synapses and from experimental data.

1.c. Spatial resolution of the morphologies used in the study

The model’s assumption that the morphology of astrocytes is tree-like is now strongly questioned in the field, with most recent reports highlighting a reticular/spongiform morphology of fine processes: see e.g Shigetomi et al., 2013, . J. Gen. Physiol.; Rusakov, 2015, Nat rev Neurosci and Arizono et al., 2020, Nature Communications. If I understood correctly, the spatial resolution of the reconstruction used to validate the modelled astrocyte morphologies is 0.51 µm, which does not allow to resolve most of astrocytic surface area (~75% according to Bindocci et al, Science, 2018). The morphologies presented in this study thus do not account for the fine astrocytic processes of the spongiform domain reported by electron microscopy and super-resolution microscopy. This, as well as the relevance of the proposed geometries, should be critically discussed in the manuscript. Previous morphological analysis of astrocyte morphology should be cited and discussed: Medvedev et al, Phil. Trans. R. Soc. B, 2014; Savtchenko et al, Nature Communications, 2018 and Minge et al, 2021, Frontiers in Cellular Neuroscience. The latter characterized the heterogeneity of astrocyte morphology in the hippocampus during development and is thus crucial to be cited and discussed here.. In any case, the conclusion at the end of the introduction “astrocyte tips […] represent a small proportion of the entire astrocytic membrane” and the prediction that most astrocyte surface is contacted by other compartments than fine processes is against recent reports in the field.

2. Manuscript organization

While the text is well written, taken aside few typos such as “this findings”, the manuscript lacks conciseness and a logical framework. Conclusions vary between the abstract/introduction/“main points”/conclusion, so that the main insights from the paper are unclear.

The manuscript would gain in clarity with a bit of reorganization. Many redundancies are observed, notably between the Methods and Results sections. Citation of figures goes back and forth. To improve the readability of the manuscript, I would suggest to move part of the calculations that are not delivering the main messages of the paper as supporting information (for example Figures 5-9). The introduction, while showing a deep understanding of the physiology of glutamate transporters, contains non-essential information for this study and could be shortened to improve readability.

Most morphological data are presented within tables (e.g Tables 4 and 6), but would be easier to read and interpret if presented as plots.

3. Sensitivity analysis of the model predictions

The model predictions are very interesting. To what extent are they robust depending on parameter values used in the study? For example, as astrocyte-synapse distance is highly variable, to what extent does the glutamate concentration profile predicted by the model depend on this distance?

In the model, glutamate is released at the centre of the cleft rather than at the membrane of the pre-synaptic element. Does the location of glutamate release impact the model’s predictions?

The open probability of glutamate receptors are calculated using ChanneLab 2.041213. Does this method provide similar results to 3D Monte Carlo reaction-diffusion simulations that would include those receptors at the post-synaptic membrane?

4. Code availability

While the code for generating in silico astrocytes is on Github, it seems that the access is restricted so that I could not review or test the code. The availability of the code generating 3D Monte Carlo reaction-diffusion simulations was not mentioned.

MINOR POINTS

- Numbered lines would be greatly appreciated to facilitate review and comments.

- The fact that transporters account for 11-27% of all membrane proteins is surprising and requires discussion

- Summary figure: some legend is lacking (spillover/synapse). It could also be interesting to add the effect of transporter density on the extrasynaptic receptor activation as it is highlighted both in the abstract and in the “main points” section.

Reviewer #2: In the project "Estimating the glutamate transporter surface density in mouse hippocampal astrocytes" Radulescu at al., estimate the local surface density of GLAST and GLT-1 in different areas of hippocampal astrocytes in mice aged 2-3 weeks, 7-8 weeks and 15 months. The authors with experimental data, stochastic model and analytical calculations revealed that the astrocyte ends, which hold the highest density of transporters, represent a small proportion of the entire astrocytic membrane and proposed that most synapses are contacted lower transporter membrane.

From my perspective, the manuscript is a good mixture of experimental and simulation data to discover an essential parameter of extracellular glutamate concentration, namely, the spatial distribution of glutamate transporters in mouse hippocampus.

In principle, the work is essential, necessary, with exciting results. However, a significant drawback is that the manuscript is poorly organized. I got the impression that the material was collected in a hurry, leaving something unnecessary and missing something important. The manuscript consists of three practically non-overlapping parts, an experiment to estimate the density of Glu transporters, stochastic modelling of the glutamate dynamics, and an analytical estimate of the maximum density of triangles on various astrocytic surfaces. These parts don't complement each other. It is necessary to reorganize the material to clarify how these three parts complement each other.

Formally, the main idea of the manuscript can be described in the following sentence. The density of glutamate transporters in the mouse hippocampus is higher than in the rat hippocampus, and transporters are mainly not at the tips of the astrocyte (because there are few of them) but on large branches.

By the way, the main conclusion that I saw in stochastic modelling is that the higher the density of transporters, the lower the concentration of glutamate. However, this is obvious even without modelling. Thus, the reading of the manuscript raises the question of why, in principle, a model of glutamate diffusion and uptake is needed to define the density of transporters estimated from experiment and surface density calculations. Therefore, at the very beginning, in the introduction, the authors should emphasize what role of stochastic simulations of glutamate plays in determining the density and distribution of glutamate transporters.

However, for the sake of fairness, in general, the results may be of interest to a wide range of readers working in the field of neuron and astrocytes communications.

Here the questions and advice to the authors to improve the manuscript.

1. I did not find any software at this address, and even the page itself is missing https://github.com/scimemia/Astrocyte generation.

2. 2. Please use the math format for numbers (10^2 and so on), not a digital format like 1e2 or 1e^2.

3. Page.10. The authors determined the diffusion region with absorbing boundaries. However, a computation area with a fixed size is required only for a given diffusion mesh. Otherwise, if each particle is independent and has a different diffusion's step, which is very important for complex geometric regions, it makes no sense to determine the size of the calculation's area. It's just that when analyzing diffusion, you can ignore particles that have moved away from the point of release at a certain distance. In this case, there will be no problems with strange absorbing borders. The condition of absorbing boundaries creates an additional flux of these particles, which sharply decreases the concentration of ions inside the diffusion region.

4. Page. 10. «Astrocyte branch diameter generation.» The authors introduce an algorithm for generating a geometric model of astrocytes in silico. In the completed model, the average astrocyte diameter is 1 µm, which means that the surface-to-volume ratio will be 2 µm-1. Typically, however, this value is in the order of 20 μm-1, which corresponds to a diameter of 100 nm. Some comments are needed here.

5. The section "Three-dimensional modelling of the Monte Carlo diffusion-reaction" requires a more accurate description of the algorithm for the numerical interaction of glutamate molecules with transporters. Interaction is a critical point in modelling. As far as I understand, the transporters were presented in concentrations and not in individual particles? However, glutamate is represented by individual particles, which still need to be converted into concentration. It is not clear from the method how the average concentration of glutamate was calculated, over what volumes were averaged, etc. Moreover, it is necessary to say a few words why such a model and such parameters of transporters kinetics were chosen.

6. The part "Glutamate transporter spatial distribution" is interesting. The problem of the optimal arrangement of figures on the surface is complex and requires profound investigation. Furthermore, this is, of course, an essential part of the manuscript. However, proteins are incorporated into membranes randomly, so the number of randomly included particles (non-overlapping) will always be much smaller than the maximum quantity. Maybe the authors will find a method to estimate the deviation of the number of transporters at random distribution from their maximum number.

7. Fig.1 E and D. Surface density in um-2 (not um-3)

8. Page. 20 "An increase in astrocytic glutamate transporter expression did not lead to changes in the glutamate concentration profile in the synaptic cleft (Figure 3C)". It is necessary to correct this phrase since the glutamate dynamic is shown at the synaptic cleft centre, and this is why the concentration-independent of transporters. However, at the cleft edge, the concentration can vary depending on the density of transporters. In any case, that is related to my previous question, how do the authors calculate the glutamate concentration inside the cleft and outside?

9.

Reviewer #3: The study investigates the effects of varied expression of glutamate transporters on astrocytes of the mouse hippocampus, on synaptic signaling. This an important study for various reasons. The activity of transporters is critical for the specificity of signaling and preventing excitotoxicity. Consequently, several neurological disorders can be correlated with the aberrational activity of transporters. Existing data is based on studies on adult rats, however, most experiments and models use mice in various stages of development and continue to use data based on rat experiments. This study extensively reports on expression levels of transporters across age groups and localizations of transporter subtypes in different parts of the cell in a mouse using a wide variety of experimental techniques. The study then uses this data to develop physiologically realistic 3d models of the astrocytic processes and accounts for varying coverage of synapses in the hippocampus. The results quantitatively describe how changes in expression of transporters modify synaptic signaling via activation of extrasynaptic AMPA and NMDA receptors.

It was a pleasure to read the manuscript. The results are important for several future studies on synaptic signaling in the hippocampus. The methods are described with great care and detail. This manuscript is an example of how detailed biophysical modeling can provide physiological insights that can chart the course of future investigations.

**Have the authors made all data and (if applicable) computational code underlying the findings in their manuscript fully available?**

Reviewer #1: **No: **While the code for generating in silico astrocytes is on Github, it seems that the access is restricted so that it could not be reviewed or tested. The availability of the code generating 3D Monte Carlo reaction-diffusion simulations was not mentioned.

Reviewer #2: **No: **I did not find any software at this address, and even the page itself is missing https://github.com/scimemia/Astrocyte generation.

Reviewer #3: None

PLOS authors have the option to publish the peer review history of their article (what does this mean?). If published, this will include your full peer review and any attached files.

Reviewer #1: No

Reviewer #2: No

Reviewer #3: **Yes: **Suhita Nadkarni
---

## [Decision Letter · Decision Letter 1]

19 Dec 2021

Dear Dr Scimemi,

Thank you very much for submitting your manuscript "Estimating the glutamate transporter surface density in distinct sub-cellular compartments of mouse hippocampal astrocytes" for consideration at PLOS Computational Biology. As with all papers reviewed by the journal, your manuscript was reviewed by members of the editorial board and by several independent reviewers. The reviewers appreciated the attention to an important topic. Based on the reviews, we are likely to accept this manuscript for publication, providing that you modify the manuscript according to the review recommendations.

In particular, we agree with the comments of Reviewer#1 on statistical significance testing. Please ensure to address all of the points raised by this reviewer in your revision.

Sincerely,

Hugues Berry

Associate Editor

PLOS Computational Biology

Lyle Graham

Deputy Editor

PLOS Computational Biology

[LINK]

Reviewer's Responses to Questions

**Comments to the Authors:**

Reviewer #1: The authors have addressed my previous comments. A lot of additional work and experiments/simulations have been performed and the manuscript has greatly improved. My previous comments have been carefully read and answered.

I however still have some major concerns regarding the manuscript in its current form:

- My main concern probably resides in the lack of any statistical analysis in the paper. For example, similarity between the results obtained in the study and other works is often asserted without any justification/quantification (e.g considering that 65% is “close enough” to 75%). Further, no analysis is presented to quantify the similarity between the properties of the model and the experimental measurements. Accordingly, there is no Data analysis/Statistical analysis section in the Methods section. Overall, the manuscript would greatly improve by using data analysis techniques, describing and justifying them (see Makin & Orban de Xivry, 2019, eLife, DOI: 10.7554/eLife.48175).

- Although all the model and calculation assumptions are now clarified in the text, some of the assumptions are hard to understand. For example, the authors assert that the proportion of GLT-1 expressed by neurons is negligible while stating that they account for 5-10% of its total expression in the brain (which is not negligible). Why not including those 5-10% in the calculations rather than assuming that all GLT-1 is astroglial?

- I have concerns regarding the quality of the reconstruction of the astrocyte leaves in Figure 2. The resolution of the method should be clearly stated and comparison with shapes/morphologies previously obtained by other groups should be mentioned. Further, if I understood correctly, it seems that any non-neuronal element near the synapse was considered being an astrocyte.

- Conclusions are often too assertive. For example, the results from this study suggest but do not show (l.39) that there are stark differences in transporter density depending on the subcellular compartment. Another example: the authors state that the density of the transporters decrease with age while, in Fig5, the decrease of surface density at 15 months is followed by an increase at 21 months old. This latter is however surprising and would be interesting to discuss. Finally, the speculations regarding the 3/2 rule applicable to astrocytes (l599-603) are way too assertive compared to the degree to which it was proven in the paper.

- Some of the results are surprising and require more discussion. For example, l363-366, the authors suggest that at 6 months old, 100% of the astrocyte surface area in leaves would be occupied by transporters. This is however highly unlikely as astrocytes express other proteins in their PAPs (see Mazaré et al, 2020, Cell Reports), including some located at the plasma membrane.

- The choice of plots to present the data is sometimes misleading, notably in the section presenting the modeling results. As reaction-diffusion simulations are performed, I do not understand why Fig 6F,H and J are presented with lines rather than scatter/bar plots. Error bars are also lacking. Importantly, the right panels from Fig8 are misleading and give the impression that simulations were performed with numerous values of astrocyte-release site distance, while only 3 values were tested (close, far, center). Here also, the variability depending on seed value (error bar/STD) should be added to the plots (if the model is indeed a reaction-diffusion model, see also next point).

- Information on the model is lacking, such as parameter values (other than geometric parameter values) and the associated references. Presenting them in a schematic as well as a table, for example, is crucial for reproducibility purposes. Further, the details of the modeling technique used are still unclear to me. Indeed, the authors state that they are performing reaction-diffusion simulations while it seems that the model is in fact compartmental (divided into 2 micrometer cube cubes according to Supplemental Fig S6?). Could it be possible to clarify that in the Methods section?

- A lot of the conditions chosen to create the astrocyte morphologies still seem arbitrary (example: the length, orientation, distribution of the leaves)

Minor concerns:

- The reason why the model contains 7 synapses, while glutamate spillover seems to be only monitored at one synapse, is unclear.

- The term ‘clustering’ should be defined and used carefully, as it commonly refers to non-homogeneous distributions of transporters within nanodomains. In this study, the transporters seem to be heterogeneously distributed onto the surface of the astrocyte.

- Calling the extracellular space outside the cleft “extra-cellular space” is misleading as the cleft itself also consists in extracellular space.

- Most of the new paragraphs in the introduction section discuss the choices made during the study and the approximations and calculations (l66-100, 101-109, l111-129). The manuscript would probably gain in clarity if this text was transferred to the Discussion & Supplemental sections.

- l151-162 + Fig1 probably belong to the Methods section.

- Definition of the terminology, notably the definition of shafts vs leaves, is sometimes lacking. This is crucial as those terms can mean different things depending on the laboratory (it would be nice if the community reaches a consensus on this terminology!)

- Fig7: why testing only 3 data points (close, center, far)? At what time is the probability of occupancy computed? Is it the average from the whole simulation time?

- Fig 7 and 8 could probably be combined as they investigate the same question.

- How do the proposed morphologies compare to the in silico astrocyte morphologies provided by e.g. Savtchenko et al, Nat. Neuroscience, 2018?

- l460-461: part of the sentence is missing

- The blender files lack instructions to allow other users to be able to smoothly reuse the code. The required software and repositories to run the code should be clearly listed in the README file.

Reviewer #2: The manuscript “Estimating the glutamate transporter surface density in mouse hippocampal astrocytes" by Radulescu et al. has been improved, and it is easier to follow the logic.

The result is of interest to specialists in the field of membrane transporters. The goal of the ms intends to reveal the difference between the density of transporters in the membranes of rat and mouse astrocytes.

I have no technical objection to publishing the results.

Reviewer #3: I have no further concerns

**Have the authors made all data and (if applicable) computational code underlying the findings in their manuscript fully available?**

Reviewer #1: None

Reviewer #2: Yes

Reviewer #3: None

PLOS authors have the option to publish the peer review history of their article (what does this mean?). If published, this will include your full peer review and any attached files.

Reviewer #1: No

Reviewer #2: No

Reviewer #3: **Yes: **Suhita Nadkarni

Figure Files:

Data Requirements:

Reproducibility:

References:

---

## [Decision Letter · Decision Letter 2]

18 Jan 2022

Dear Dr Scimemi,

We are pleased to inform you that your manuscript 'Estimating the glutamate transporter surface density in distinct sub-cellular compartments of mouse hippocampal astrocytes' has been provisionally accepted for publication in PLOS Computational Biology.

Best regards,

Hugues Berry

Associate Editor

PLOS Computational Biology

Lyle Graham

Deputy Editor

PLOS Computational Biology

Reviewer's Responses to Questions

**Comments to the Authors:**

Reviewer #1: The authors have addressed all of my previous concerns and the manuscript has improved greatly. I have few comments, listed below, although I believe that the paper is now ready for publication.

- The majority of PAPs are believed to belong to the leaflets category (Aboufares El Alaoui et al, 2021, Frontiers in Cellular Neuroscience, DOI: 10.3389/fncel.2020.573944). Further, even if leaves/tips were distinct from PAPs, the probability that all molecules on their surface are GLT-1 transporters is still very low.

- Figures 6 and 8: it could be helpful to the reader if the legend specified that the lines facilitate the interpretation of the results but do not correspond to simulation results.

**Have the authors made all data and (if applicable) computational code underlying the findings in their manuscript fully available?**

Reviewer #1: Yes

PLOS authors have the option to publish the peer review history of their article (what does this mean?). If published, this will include your full peer review and any attached files.

Reviewer #1: No

---

## [Editor Report · Acceptance letter]

1 Feb 2022

PCOMPBIOL-D-21-00964R2 

Estimating the glutamate transporter surface density in distinct sub-cellular compartments of mouse hippocampal astrocytes

Dear Dr Scimemi,

I am pleased to inform you that your manuscript has been formally accepted for publication in PLOS Computational Biology. Your manuscript is now with our production department and you will be notified of the publication date in due course.

With kind regards,

Agnes Pap
